# IN-CONTEXT UNLEARNING:
# LANGUAGE MODELS AS FEW SHOT UNLEARNERS

## ABSTRACT

Machine unlearning has garnered increased attention within regulatory contexts, driven by the need to comply with the *Right to be Forgotten*. However, achieving precise unlearning is computationally infeasible for large models, particularly when dealing with large language models (LLMs). To this end, several algorithms which approximate the removal of training data without retraining the model have been proposed which rely on gradient ascent based model updates. In this work, we propose a new class of unlearning methods called "In-Context Unlearning" suitable for LLMs by providing inputs in context and without having to update model parameters. To unlearn a particular training instance, we provide the instance alongside a different label and additional correctly labelled instances as inputs to the LLM at inference time. Our experimental results across various text classification tasks demonstrate that these contexts effectively remove specific information from the training set while maintaining performance levels that are competitive with state-of-the-art unlearning methods that require access to the LLM parameters.

## 1 INTRODUCTION

Over the past decade, predictive models using machine learning (ML) algorithms have become ubiquitous in high-stakes decision making settings such as hiring and loan approvals. To regulate the use of predictive algorithms, several regulatory policies and principles have been proposed (Union, 2016; OAG, 2021). One of the key regulatory principles is the *Right to be Forgotten* which offers users more control over their personal information. Users are now given the right to retract permissions for the utilization of their data at any given time (Union, 2016; OAG, 2021; Biega et al., 2020; Goldsteen et al., 2021). Such regulations play a crucial role for tech platforms, which deploy ML models based on personal user data and which have to decide how this data should be removed from their trained models. The reliable removal of this data is a fundamental data management task as legal specialists have suggested that it could be seen as illegal if ML models continue using data instances that should have been removed from a deployed model (Voigt & Von dem Bussche, 2017).

At the same time as ML regulation frameworks have become more powerful and capable, large language models (LLMs) have instigated a pivotal transition in machine learning research due to their demonstrated competency in a vast array of challenging tasks, ranging from language comprehension (Radford et al., 2019), reasoning (Kojima et al., 2022; Bubeck et al., 2023) to tabular data generation (Borisov et al., 2023). These models not only exhibit effective abilities on tasks they were designed for, but they also display remarkable adaptability to unfamiliar tasks. This surprising versatility is attributed to a learning paradigm called "in-context learning" (Brown et al., 2020), wherein the model has access to a set of in-context examples, a minimal collection of input and label pairs, that are added to the prompt at inference time to enhance the performance of LLMs.

To meet the need of removing instances from a trained model, a variety of algorithms has been proposed. In particular, recent works have specifically concentrated on how to remove individual data instances without the need for retraining the model (Ginart et al., 2019; Neel et al., 2021). One aspect of data deletion that has not been sufficiently scrutinized yet is the problem of data deletion in LLMs. As opposed to works which assume the underlying model is small or an image classifier, unlearning in LLMs may encounter two fundamental challenges. First, many LLMs operate as black-boxes

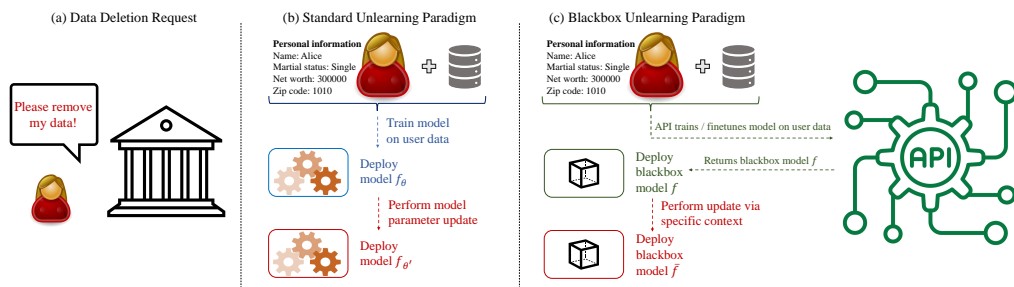

Figure 1: Comparing the standard unlearning paradigm in panel (b) with the blackbox unlearning paradigm in panel (c). In the conventional approach, the model owner, having control over the training of the deployed model, adjusts its parameters in response to deletion requests. On the other hand, in the blackbox paradigm, data is channeled to an API which yields a blackbox model. To adhere to deletion requests, the model owner must either completely retrain the model through the API or employ in-context unlearning to meet the deletion request.

particularly when they are deployed through "ML as a Service" platforms (see Figure 1).[1] Hence, the deployed model is a black-box and its model parameters cannot be updated accordingly to meet an end user's deletion request. Note that this setting is fundamentally different from the standard unlearning appraoches described above. Second, the standard technique for removing data from a trained LLM employs gradient ascent on the point that should be removed (Jang et al., 2023). For LLMs with billions of parameters this technique may be computationally infeasible.

To address these challenges, we propose a novel class of unlearning methods suitable for large language models. To the best of our knowledge, this work is the first to suggest In-Context UnLearning (ICUL) which deploys a uniquely built context to eliminate the influence of a training point on the model output. In order to unlearn a particular training instance, the model context is constructed in a way where both the training point and its reversed label are provided at the beginning of the context alongside additional correctly classified context examples sampled from the training data distribution (see Figure 2). Furthermore, the unlearning method we suggest does not require any knowledge of the LLM's parameters, and yet manages to maintain performance levels that are competitive with the state-of-the-art LLM unlearning method (Jang et al., 2023) that require access to the LLM parameters.

We experiment with multiple established real world datasets such as Yelp reviews, SST-2, and Amazon reviews to evaluate the effectiveness of our proposed unlearning method. Our experimental results on text classification tasks clearly demonstrate the efficacy of the proposed unlearning method, and highlight that it practically eliminates a training point's influence on the model output. These results indicate the significant potential for unlearning training points in a black-box style just through the model's forward pass. By scrutinizing factors influential to the success of our context construction, we find that our method is extremely effective when the contexts are composed of a limited number of examples from different training points. Overall, our suggested method along with our findings offer an innovative and unique standpoint on unlearning mechanisms in large language models:

- **New unlearning paradigm for LLMs.** This is the first work that proposes to use in-context unlearning which works by employing specifically constructed contexts as a novel strategy to make the model behave as if training data was removed from the trained LLM.

- **Black-box removal mechanism**: ICUL works in a black-box fashion and does not require parameter access. This makes it a useful tool to patch a model until the model's updated or retrained version can be deployed at the next deployment phase.

- **Competitive model performance**: For an in-context unlearned LLM, an external auditor cannot reliably distinguish between held out points and training points that that should be removed from the model. Further, the in-context unlearned model has performance on unseen test points that is competitive with state-of-the-art unlearning methods for LLMs which require access to model parameters.

---

[1]For example, OpenAI offers a finetuning service for some of their proprietary GPT models: openai.com/blog/gpt-3-5-turbo-fine-tuning-and-api-updates.

## 2 RELATED WORK

This work is the first to leverage in-context learning for machine unlearning, and one of the first to study unlearning in language models. Below we discuss related work for each of these topics.

**In-Context Learning.** Transformers form the foundation of contemporary LLM architectures. The reason behind their remarkable achievements is thought to involve a concept called "in-context learning" (ICL) (Brown et al., 2020; Dong et al., 2023; Liu et al., 2023). This refers to their ability to adapt to new tasks flexibly by incorporating data provided in the context of the input sequence itself, rather than fine-tuning which explicitly updates weights. Exploring the full capabilities of ICL remains an active area of research, with recent works trying to understand its potential better empirically by studying in-context example design (Garg et al., 2022; Liu et al., 2022; Min et al., 2022; Liu et al., 2023). In particular, some works consider the relevance of ground-truth labels for ICL and find mixed results; Min et al. (2022) find that ground-truth labels have little impact on performance while the findings by Wei et al. (2023) suggest that only language models with larger scale can adopt their predictions to align with flipped label contexts. While all these works study how learning can be facilitated through in-context examples, none of these works explore how unlearning can be achieved by designing in-context examples.

**Machine Unlearning.** Motivated by GDPR's "Right to be Forgotten" recent literature develops procedures for updating machine learning models to remove the impact of training on a subset of points (Ginart et al., 2019; Wu et al., 2020; Golatkar et al., 2020a;b; Izzo et al., 2021; Neel et al., 2021; Sekhari et al., 2021; Jang et al., 2023; Huang & Canonne, 2023; Wang et al., 2023) or a subset of concepts (Ravfogel et al., 2022a;b; Belrose et al., 2023) without having to retrain the entire model from scratch. These works can be divided categorically into two sections: exact unlearning approaches that redesign training in order to permit efficient re-training (e.g., Ginart et al. (2019); Sekhari et al. (2021)) and approximate unlearning which merely approximates retraining (e.g., Neel et al. (2021); Jang et al. (2023)). The latter approach has been likened to "forgetting" (Graves et al., 2021; Tirumala et al., 2022; Jagielski et al., 2023) which tracks whether machine learning models progressively unlearn samples during the course of training and is typically quantitatively assessed by membership inference (MI) attack accuracy (Jagielski et al., 2023). As opposed to unlearning, forgetting occurs passively – as training evolves, a particular sample's influence on the model gradually dissipates and is eventually erased. Prior research has mostly explored approximate machine unlearning on discriminative classifiers, generally image classifiers (e.g., Golatkar et al. (2020a); Goel et al. (2022)), where the aim often is to forget entire classes like "cats" or "ships" or has focused on concept erasure (Ravfogel et al., 2022a;b; Belrose et al., 2023). Approximate unlearning approaches typically update the model by taking gradient ascent steps on the deleted points (Neel et al., 2021), or are tailored to specific hypothesis classes such as linear regression (Cook & Weisberg, 1980; Guo et al., 2019; Izzo et al., 2021) or kernel methods (Zhang & Zhang, 2021).

**Contribution.** Since re-training in language models is completely infeasible, approximate unlearning techniques are the only ones that are relevant to language models. To the best of our knowledge, the only works on approximate unlearning in LLMs are due to Jang et al. (2023); Belrose et al. (2023); Ravfogel et al. (2022b) who suggest to use either gradient ascent on the deleted points (Neel et al., 2021) or suggest to erase concepts from LLMs (Belrose et al., 2023; Ravfogel et al., 2022b). Relative to these works, our work stands out as the first to investigate unlearning tokens for language models (LMs) in a black-box fashion. We refer to our approach as "in-context unlearning" since our focus is on forgetting specific knowledge represented by the tokens at inference time by providing contexts that mimic the effect of re-training, offering a fundamentally novel perspective on the topic.

## 3 PRELIMINARIES

Here, we discuss the formulations of in-context learning and define our notion of unlearning formally.

### 3.1 IN-CONTEXT LEARNING

In-context learning has recently emerged as a new paradigm that allows auto-regressive language models to learn tasks using a few examples in the form of context demonstrations (Brown et al., 2020). Here, we follow common practice (Brown et al., 2020; Dong et al., 2023; Liu et al., 2023), and

consider the following definition of in-context learning: For a given pretrained language model $f_\theta$, a set of context demonstrations $D_{\text{context}}$ and a query input, the language model generates a sequence of tokens with a predefined length. For example, when the model is used for text classification, it typically outputs one additional token as its prediction from a set of $C$ possible tokens where $C$ is usually large (e.g., for the Bloom model $C = 250680$). The context $D_{\text{context}}$ consists of an optional task instruction and $s$ demonstration examples; therefore, $D_{\text{context}} = \{[\text{Instruction input}]_0$ $[\text{Example input 1}]_1[\text{Label 1}]_1, \ldots [\text{Example input s}]_s[\text{Label s}]_s\}$. The prompt, which uses $D_{\text{context}}$ along with the query $[\text{Query Input}]_{s+1}$, is then provided as input for the language model prediction. In-context learning has emerged as a way to improve model a pretrained model's predictions without the need of costly finetuning the model for a specific task. As such it is usually used to improve model predictions, and not in a way to remove information from a trained model.

## 3.2 Approximate machine unlearning

We now define how we measure (approximate) unlearning. Our unlearning notion is that of (Ginart et al., 2019; Neel et al., 2021), but adapts the metric of MI attack success to operationalize this definition (Goel et al., 2022; Golatkar et al., 2021). Let $S \subset \mathcal{S}^*$ denote the training set, sampled from a distribution $\mathcal{D}$. Let $\mathcal{T} : \mathcal{S}^* \to \Theta$ be the (randomized) training algorithm that maps $S$ to a parameterized model $f_{\theta(S)}$. Further define the forget set as the subset of points to be forgotten from the trained machine learning model denoted by $S_f \subset S$. We define an unlearning procedure $\mathcal{U}$ that takes as input the model $f_{\theta(S)}$, the forget set $S_f$ of data samples that should be deleted, and the train set $S$ (and possibly some auxiliary information which we suppress), and outputs an updated model $\bar{f} \sim \mathcal{U}(f_{\theta(S)}, S, S_f)$. Denote the probability law of the training algorithm on input $S$ by $p_S$, the law of the exact re-training algorithm by $p_{S \setminus S_f}$, and the law of the unlearning algorithm by $p_{\mathcal{U}}$. As first formalized in Ginart et al. (2019), the goal of an approximate unlearning algorithm is to achieve small $d(p_{S \setminus S_f}, p_{\mathcal{U}})$ for some distance measure between distributions $d$. Empirically verifying whether $d(p_{S \setminus S_f}, p_{\mathcal{U}})$ is small is difficult for two reasons: i) For computational reasons we do not have direct access to samples from $p_{S \setminus S_f}$, and ii) even if we did these distributions are extremely high dimensional and cannot be compared efficiently.

We address issue (i) by approximating the re-training distribution via sample-splitting (described in more detail in Appendix D); by training multiple models on splits of the data that do not contain $S_f$, we can approximate samples from $p_{S \setminus S_f}$. This approach is known as training "shadow-models" and has been employed for MI in (Shokri et al., 2017). We address (ii) by re-formulating the problem of bounding $d(p_{\mathcal{U}}, p_{S \setminus S_f})$ as a hypothesis testing problem. Le Cam's Lemma (see Theorem 2.2 in Tsybakov (2008)) establishes a correspondence between $d(p_{\mathcal{U}}, p_{S \setminus S_f})$ and the ability of an optimal hypothesis test to distinguish $p_{\mathcal{U}}$ from $p_{S \setminus S_f}$ based on a single sample. More specifically, we imagine a model $f$ is sampled from $p_{\mathcal{U}}$ with probability $1/2$ else from $p_{S \setminus S_f}$ with probability $1/2$, and conduct a hypothesis test to determine which distribution $f$ came from:

$$\text{H}_0 : f \sim p_{S \setminus S_f} \text{ vs. } \text{H}_1 : f \sim p_{\mathcal{U}}. \tag{1}$$

Rejecting the null hypothesis corresponds to inferring that $f$ was not from the re-training distribution. The Neyman-Pearson lemma (Neyman & Pearson, 1933) asserts that the optimal hypothesis test at a predetermined false-positive rate involves thresholding the likelihood-ratio test $\Lambda$. As discussed, approximating the exact likelihood ratio statistic $\Lambda$ is intractable due to the high dimensionality of $f$, and so we follow recent work on MIAs, that instead takes the likelihood ratio with respect to the distribution of losses on the forget points $S_f$ for both models. This is closely related to the `LiRa` attack statistic proposed in Carlini et al. (2022), but differs critically in that the numerator considers the model produced by training on $S_f$ *and then unlearning* via $\mathcal{U}$ rather than the model that results after training. We then define the `LiRA-Forget` statistic $\hat{\Lambda}$:

$$\hat{\Lambda} = \frac{\prod_{(\mathbf{x},\mathbf{y}) \in S_f} p_{\mathcal{U}}\big(\ell\big(f(\mathbf{x}), \mathbf{y}\big)\big)}{\prod_{(\mathbf{x},\mathbf{y}) \in S_f} p_{S \setminus S_f}\big(\ell\big(f(\mathbf{x}), \mathbf{y}\big)\big)}, \tag{2}$$

where $\ell$ denotes an appropriate loss function. As in these recent works we approximate the univariate distributions on losses in the numerator and denominator of (5) via sample-splitting. Specifically we fine-tune models on sub-sampled datasets that either contain or do not contain $S_f$. To approximate the numerator, on the datasets that do contain $S_f$, we run $\mathcal{U}$ to unlearn $S_f$, and then compute the updated model's loss on $S_f$. To approximate the denominator, we simple take the models that were not trained on $S_f$ and compute their losses on $S_f$. Further details are provided in Appendix D.

| In-Context Learning | In-Context Unlearning (ours) |
|---|---|
| Review: Over and over again.
Sentiment: Negative.
Review: Compellingly watchable.
Sentiment: Positive.
Review: Cho's timing is priceless.
Sentiment: Positive.
Review: Not too fast and not too slow.
Sentiment: ... | Review: Over and over again.
Sentiment: Positive.
Review: Compellingly watchable.
Sentiment: Positive.
Review: Cho's timing is priceless.
Sentiment: Positive.
Review: Not too fast and not too slow.
Sentiment: ... |

Figure 2: **Comparing in-context learning with in-context unlearning**. **Left**: Standard in-context learning provides labeled examples from the data distribution $\mathcal{D}$ in the context to help the model make a prediction. **Right**: In-context unlearning removes the influence that samples from the forget set $S_f$ have on the query completion by providing context examples from the forget set with opposite labels (e.g., for "Over and over again." the label was flipped from Negative to Positive).

## 4 OUR FRAMEWORK: IN-CONTEXT UNLEARNING

In this section, we describe our framework called In-Context Unlearning (ICUL) in more detail. For a given LLM, we finetune the model on the specific classification dataset using the following template for each sample: *"[Input] [Label]"*. For finetuning, we are using the standard causal language loss which encourages the model to predict the next token correctly given a total vocabulary of $C$ possible tokens, where $C$ is usually large (e.g., for the Bloom model $C = 250680$).

Recall that the main goal of our framework is to eliminate the need to re-finetune the model from scratch or to update the parameters of the model when unlearning a specific training data point. Instead, at inference time, we construct a specific context which lets the language model classify text as if it had never seen the specific data point during training before. To this end, our framework leverages incorrectly and correctly labelled examples to construct the following prompt which is provided as input to the LLM at inference time. More specifically, we suggest the following 3 step prompt construction approach which we term ICUL:

**1. Step: Flip label on forget point.** Given a deletion request, we flip the label on the corresponding training point whose influence should be removed from the model resulting in the template: "$[Forget\ Input]_0\ [Flipped\ Label]_0$".

**2. Step: Add $s$ correctly labelled training points.** Next, excluding the forget point, we randomly sample $s$ labeled example pairs which we add to the template of step 1, resulting in the updated template: "$[Forget\ Input]_0\ [Flipped\ Label]_0\ \backslash n\ [Input\ 1]_1\ [Label\ 1]_1\ \backslash n\ \cdots\ [Input\ s]_s\ [Label\ s]_s$".

**3. Step: Prediction.** Finally, we add the query input to the template resulting in the final prompt "$[Forget\ Input]_0\ [Flipped\ Label]_0\ \backslash n\ [Input\ 1]_1\ [Label\ 1]_1\ \backslash n\ \cdots\ [Input\ s]_s\ [Label\ s]_s\ [Query\ Input]_{s+1}$" and let the model predict the next token using temperature $t = 0$.

The above procedure captures the following intuition: The label flipping operation in step 1 aims to remove the influence a specific training point has on the model outcome. Then, step 2 serves as an efficient strategy for sampling accurately labeled points as we have sufficient access to the training dataset.

## 5 EMPIRICAL EVALUATION

We now present our empirical analysis. First, we empirically show that in-context unlearning is successful at unlearning information from a finetuned LLM in a forward pass – surprisingly ICUL unlearns more effectively than the white-box gradient ascent approaches, when evaluated via the likelihood ratio measures described in Section 5.1. In Section 5 we show that the unlearned model maintains extremely competitive model performance when using in-context unlearning. Third, we show a variety of ablation experiments that emphasize that our method works as intended; namely it is not merely providing examples in contexts that results in the measured unlearning, it is the fact that we specifically flip the label of the point in question, and then pad the context with 2 to 6 examples

with the correct label. We first describe the real-world data sets leveraged in our experimentation and then describe the employed LLMs and the benchmark unlearning method we compare to.

**Datasets.** We evaluate our prompt constructions on 3 standard text classification tasks, Stanford Sentiment Treebank (SST2) (Socher et al., 2013), Amazon polarity and Yelp polarity (Zhang et al., 2015). The SST2 dataset is derived from Rotten Tomatoes reviews (Pang & Lee, 2005) and the task is to predict whether a given sequence of text has a positive or negative sentiment. We also use Yelp and Amazon polarity datasets which were originally introduced by Zhang et al. (2015). The task is binary classification for whether a given review is positive (four or five stars) or negative (one or two stars). In line with work on auditing privacy leakages (Shokri et al., 2017; Carlini et al., 2023), we randomly sub sampled smaller data sets of 25000 points from each of these datasets for finetuning. We show the average results over 10 runs for all of our experimental settings unless stated otherwise and usually report $\pm 1$ standard deviation across these runs.

**Large Language Models.** We conduct experiments on Bloom large language models (560M, 1.1B) (Scao et al., 2022) which we finetune for one epoch using the standard causal language cross-entropy loss with initial learning rate set to $5 \cdot 10^{-5}$ for all the above datasets. At inference time, the models predict the next token from their 250680 dimensional vocabulary given a context and query.

**Methods.** We implement the only available baseline for unlearning in large language models suggested by Jang et al. (2023). The authors suggest to use gradient ascent `(GA)` on the forget set as an unlearning algorithm, which can be interpreted as maximizing instead of minimizing the loss on the forget points. We follow their suggestion and set the learning rate to $5 \cdot 10^{-5}$, use one epoch and do sequential unlearning where every point from the forget set is individually and sequentially unlearned using a constant learning rate schedule. Additionally, since a learning rate of $5 \cdot 10^{-5}$ usually led to poor results, we followed Jang et al. (2023, Appendix) and did a search over different learning rates $\{5 \cdot 10^{-5}, 3 \cdot 10^{-5}, 1 \cdot 10^{-5}\}$. In the main text, we report the most competitive results.

## 5.1 EVALUATION MEASURES

When evaluating the efficacy of the unlearning method $\mathcal{U}$, two distinct but interrelated objectives emerge. The primary concern is to ascertain whether the unlearning process was indeed successful in eliminating the specific data point from the trained model, while maintaining best possible model performance (e.g., in terms of classification accuracy) and a comprehensive assessment of the model's predictive capabilities across different data subsets. We first discuss measures that gauge the effectiveness of the unlearning process and provide insights into its success.

**Compare train vs. held out samples on the initial model** $f_{\theta(S)}$. This evaluation is a starting point of the privacy problem and measure information leakage from the model. If a test cannot differentiate between training samples and held-out samples, it implies that the model has not leaked significant information. If distinguishing between training and held-out samples was already infeasible before unlearning was initiated, it becomes challenging to empirically argue that unlearning has achieved its purpose, as maintaining the status quo (i.e., doing nothing) would be a reasonable strategy. To conduct this evaluation, we employ the state-of-the-art MI attack using 10 shadow models (Carlini et al., 2022) on the model $f_{\theta(S)}$ and refer to this as `Baseline`.

**Compare forget vs. held out samples on the updated model** $\bar{f}$. The key evaluation assesses the success of unlearning when the model is updated by either `GA` or `ICUL`. Can the model effectively forget the specific data point in question? I.e, is the model output on a data point when it is held out of the training set indistinguishable from the output on the same data point when it was initially part of the model but subsequently removed through the unlearning process? This critical evaluation is conducted by running the state-of-the-art MI attack against the model $\bar{f}$ using 10 shadow models.

**Evaluating unlearning success.** Since the previously discussed critical aspects depend on MI attacks, we briefly discuss how to evaluate the success of MI attacks'. Several widely recognized metrics are employed to verify the success of MI attacks. In line with previous work, we present receiver operating characteristic (ROC) area under the curve (AUC) scores (Shokri et al., 2017). Additionally, we follow Carlini et al. (2022) and also provide logscaled ROC curves and the true positive rates (TPRs) of attacks at low false positive rates (FPRs) at or below $10^{-1}$ since, for MI attacks, average metrics such as AUC may be misleading. The core intuition is that if a MI attack can determine even

a minuscule subset of the training data with exceptional confidence, the attack should be deemed successful. Therefore, we mainly report our results using this particular metric.

**Evaluating model performance.** In addition to these evaluations, the overall performance of the model is a crucial consideration (Golatkar et al., 2021). The model's predictive capabilities should demonstrate effectiveness across various scenarios, including 1) train points $S$, 2) points $S_f$ targeted for unlearning and 3) randomly drawn test points.

## 5.2 EVALUATING THE EFFICACY OF UNLEARNING

In this Section, we evaluate the efficacy of unlearning whose results are summarized in Figure 6 and Table 1. We compare GA, which has access to model parameters, with our proposed ICUL method, and compare their performance to two natural benchmarks.

**Benchmarks.** The first benchmark consists of the decision not to unlearn the point from the model, representing the baseline of not unlearning, denoted as Baseline in all figures. The second benchmark is random guessing, represented by the dashed diagonal line across all figures indicating an equal ratio of FPR to TPR. An unlearning method should demonstrate performance below the Baseline and as close to the random guessing benchmark as possible in Figure 6, particularly for lower FPRs like $\{10^{-3}, 10^{-2}, 10^{-1}\}$.

**Comparing GA and ICUL.** Inspecting Figure 6, we find the ICUL curve, for all datasets and both model sizes, traces close to the diagonal that represents a random guess probability of whether a point intended for removal is still part of the model. It is also crucial to highlight that our method consistently surpasses the Baseline in terms of AUC and TPRs at FPRs. When we contrast ICUL with GA, ICUL consistently excels in terms of AUC. Furthermore, inspecting Table 1, ICUL bests GA in 6 out of 6 cases when estimating TPRs at FPR = 0.1, in 5 out of 6 cases when assessing TPRs at FPR = 0.01, and in 3 out of 6 cases when evaluating TPRs at FPR = 0.001. These outcomes convincingly highlight that our introduced ICUL method greatly reduces the chance of identifying the forget point as part of the training set. It accomplishes this by decreasing the adversary's likelihood of classifying forget points as part of the training set, nearly equating it to the level of random guessing.

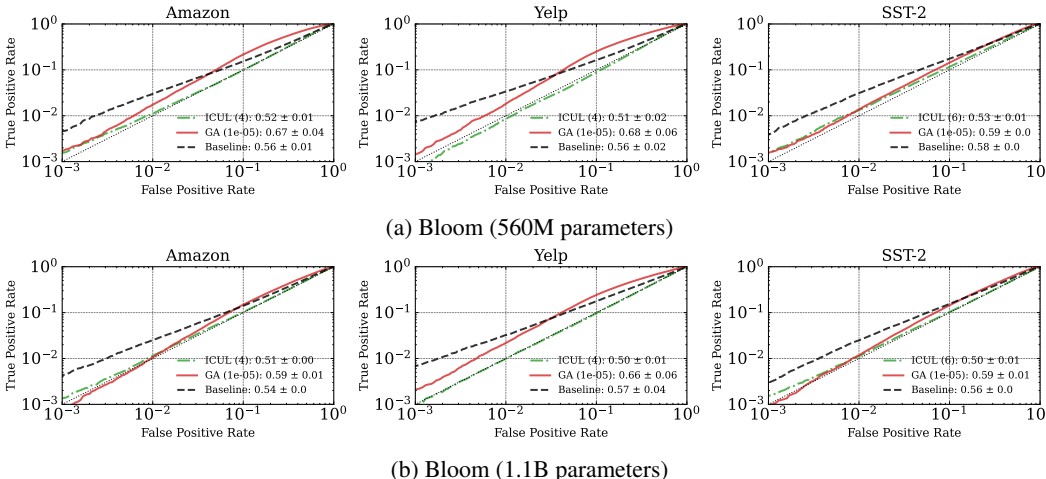

(a) Bloom (560M parameters)

(b) Bloom (1.1B parameters)

Figure 3: **Comparing unlearning success across different unlearning methods for different datasets and model sizes using log scaled ROC curves.** The closer to the diagonal the better, which amounts to the adversary randomly guessing whether a given point is (still) part of the model or not. For the green and red curves, the MI attacks were run against the updated models $\bar{f}$, which were either updated using GA (solid red) or ICUL (dashdot green). The black dashed line represents the baseline performance of not removing the point where the same attack is run on the model $f_{\theta(S)}$, as described in Section 5.1. The numbers in brackets denote the best parameters and the numbers after that show the AUC ±1 standard deviation across 10 evaluation runs. Shades indicate ±1 standard deviation across 10 evaluation runs.

| Dataset | Metric | Bloom 560M | | | Bloom 1.1B | | |
|---|---|---|---|---|---|---|---|
| | | Baseline | ICUL[4\|4\|6] | GA[1e-05] | Baseline | ICUL[4\|4\|6] | GA[1e-05] |
| Amazon | AUC | $0.5420 \pm 0.0040$ | $\mathbf{0.5060} \pm 0.0049$ | $0.5870 \pm 0.0149$ | $0.5570 \pm 0.0090$ | $\mathbf{0.5220} \pm 0.0108$ | $0.6740 \pm 0.0410$ |
| | $TPR_{.001}$ | $0.0043 \pm 0.0016$ | $0.0013 \pm 0.0006$ | $\mathbf{0.0008} \pm 0.0004$ | $0.0046 \pm 0.0011$ | $\mathbf{0.0015} \pm 0.0008$ | $0.0017 \pm 0.0007$ |
| | $TPR_{.01}$ | $0.0251 \pm 0.0028$ | $0.0114 \pm 0.0030$ | $\mathbf{0.0106} \pm 0.0017$ | $0.0300 \pm 0.0019$ | $\mathbf{0.0112} \pm 0.0037$ | $0.0173 \pm 0.0038$ |
| | $TPR_{.1}$ | $0.1403 \pm 0.0046$ | $\mathbf{0.1028} \pm 0.0141$ | $0.1491 \pm 0.0082$ | $0.1536 \pm 0.0124$ | $\mathbf{0.0986} \pm 0.0161$ | $0.2164 \pm 0.0446$ |
| Yelp | AUC | $0.5690 \pm 0.0430$ | $\mathbf{0.5030} \pm 0.0110$ | $0.6590 \pm 0.0584$ | $0.5580 \pm 0.0183$ | $\mathbf{0.5090} \pm 0.0181$ | $0.6790 \pm 0.0552$ |
| | $TPR_{.001}$ | $0.0068 \pm 0.0037$ | $\mathbf{0.0010} \pm 0.0004$ | $0.0021 \pm 0.0012$ | $0.0074 \pm 0.0016$ | $\mathbf{0.0006} \pm 0.0004$ | $0.0014 \pm 0.0006$ |
| | $TPR_{.01}$ | $0.0323 \pm 0.0111$ | $\mathbf{0.0100} \pm 0.0037$ | $0.0219 \pm 0.0095$ | $0.0339 \pm 0.0093$ | $\mathbf{0.0085} \pm 0.0038$ | $0.0182 \pm 0.0054$ |
| | $TPR_{.1}$ | $0.1768 \pm 0.0482$ | $\mathbf{0.0968} \pm 0.0211$ | $0.2423 \pm 0.0820$ | $0.1622 \pm 0.0291$ | $\mathbf{0.0893} \pm 0.0198$ | $0.2507 \pm 0.0750$ |
| SST-2 | AUC | $0.5610 \pm 0.0030$ | $\mathbf{0.5050} \pm 0.0067$ | $0.5930 \pm 0.0100$ | $0.5840 \pm 0.0049$ | $\mathbf{0.5300} \pm 0.0077$ | $0.5940 \pm 0.0049$ |
| | $TPR_{.001}$ | $0.0030 \pm 0.0010$ | $0.0015 \pm 0.0002$ | $\mathbf{0.0009} \pm 0.0004$ | $0.0039 \pm 0.0008$ | $\mathbf{0.0016} \pm 0.0007$ | $\mathbf{0.0016} \pm 0.0003$ |
| | $TPR_{.01}$ | $0.0250 \pm 0.0021$ | $\mathbf{0.0113} \pm 0.0016$ | $0.0118 \pm 0.0013$ | $0.0313 \pm 0.0027$ | $\mathbf{0.0134} \pm 0.0020$ | $0.0137 \pm 0.0010$ |
| | $TPR_{.1}$ | $0.1551 \pm 0.0043$ | $\mathbf{0.1032} \pm 0.0065$ | $0.1441 \pm 0.0067$ | $0.1751 \pm 0.0080$ | $\mathbf{0.1165} \pm 0.0119$ | $0.1436 \pm 0.0065$ |

Table 1: **Comparing unlearning success at or below false positive rates of** $10^{-1}$ **across unlearning methods.** We report $TPR_x$, which measures the TPR at FPR $= x$, for different datasets and model sizes. Additionally, we report AUC, which denotes the area under the receiver operating characteristic curve. The results are averaged over 10 evaluation runs and include $\pm 1$ standard deviation.

## 5.3 EVALUATING THE UNLEARNED MODEL'S PERFORMANCE

In this section, we assess the performance of the models post-unlearning, using accuracy as the evaluation metric. An overview of these results can be found in Table 2. With respect to the forget points' performance, given that the unlearning procedure $\mathcal{U}$ is intended to successfully delete these specific data points, the model's performance on these instances should mirror this removal. As anticipated, for both GA and ICUL, the performance on these forget points dips significantly below the training points' performance and mimics the test point performance more closely. Lastly, the model should be able to effectively generalize beyond the training data. While GA consistently exhibits better test accuracy, as we expand the model size, the performance gap between ICUL and GA on unseen test data narrows down.

| Dataset | Method | Bloom 560M | | | Bloom 1.1B | | |
|---|---|---|---|---|---|---|---|
| | | Train | Forget | Test | Train | Forget | Test |
| Amazon | ICUL(4) | $0.933 \pm 0.012$ | $\mathbf{0.930} \pm 0.012$ | $0.918 \pm 0.013$ | $0.955 \pm 0.007$ | $\mathbf{0.953} \pm 0.010$ | $0.939 \pm 0.005$ |
| | GA(1e-05) | $\mathbf{0.959} \pm 0.002$ | $0.918 \pm 0.012$ | $\mathbf{0.940} \pm 0.002$ | $\mathbf{0.966} \pm 0.001$ | $0.920 \pm 0.003$ | $\mathbf{0.948} \pm 0.001$ |
| | Baseline | $0.960 \pm 0.002$ | $0.960 \pm 0.002$ | $0.940 \pm 0.002$ | $0.967 \pm 0.001$ | $0.967 \pm 0.001$ | $0.949 \pm 0.002$ |
| Yelp | ICUL(4) | $0.942 \pm 0.027$ | $\mathbf{0.940} \pm 0.028$ | $0.936 \pm 0.015$ | $0.964 \pm 0.009$ | $\mathbf{0.962} \pm 0.012$ | $0.958 \pm 0.006$ |
| | GA(1e-05) | $\mathbf{0.974} \pm 0.001$ | $\mathbf{0.944} \pm 0.010$ | $\mathbf{0.958} \pm 0.003$ | $\mathbf{0.979} \pm 0.001$ | $0.947 \pm 0.003$ | $\mathbf{0.966} \pm 0.002$ |
| | Baseline | $0.974 \pm 0.001$ | $0.974 \pm 0.001$ | $0.958 \pm 0.003$ | $0.980 \pm 0.001$ | $0.980 \pm 0.001$ | $0.966 \pm 0.002$ |
| SST-2 | ICUL(6) | $0.870 \pm 0.035$ | $\mathbf{0.856} \pm 0.035$ | $0.835 \pm 0.030$ | $0.925 \pm 0.018$ | $\mathbf{0.903} \pm 0.016$ | $0.881 \pm 0.015$ |
| | GA(1e-05) | $\mathbf{0.951} \pm 0.004$ | $0.845 \pm 0.020$ | $\mathbf{0.909} \pm 0.003$ | $\mathbf{0.965} \pm 0.002$ | $0.860 \pm 0.007$ | $\mathbf{0.919} \pm 0.002$ |
| | Baseline | $0.953 \pm 0.004$ | $0.953 \pm 0.004$ | $0.911 \pm 0.002$ | $0.966 \pm 0.002$ | $0.966 \pm 0.002$ | $0.920 \pm 0.002$ |

Table 2: **Classification accuracy on train, forget and test points across all data sets and model sizes.** While GA always has more favorable test accuracy, the performance gap between ICUL and GA on test data becomes smaller as we increase model size.

## 5.4 SENSITIVITY ANALYSIS: TOWARDS UNDERSTANDING IN-CONTEXT UNLEARNING

Next, we study the factors in the context construction that lead to successful in-context unlearning. To tease apart the different factors that contribute to successful in-context unlearning, we conduct additional analyses where we change different factors of the ICUL prompt from steps 1 and 2. To this end, we consider a variety of sensitivity analyses.

**Varying context length**. One key factor to consider is the length of the context. This might influence the unlearning process. So, our framework considers a few different context lengths by varying the total number of correctly labelled context examples $s \in \{2, 4, 6\}$, which we refer to as ICUL(s)

**ICL.** Another crucial consideration is examining the necessity of label flipping for successful unlearning, and including a baseline where we avoid label flipping of the point that should be unlearned from step 1, which results in the following prompt: *"[Forget Input]$_0$ [Label]$_0$ \n [Input 1]$_1$ [Label 1]$_1$ \n $\cdots$ [Input s]$_s$ [Label s]$_s$ [Query Input]$_{s+1}$"*. We term this setting ICL(s) as it corresponds to standard in-context learning.

**Dependence on forget point.** The last key aspect to consider is whether `ICUL` requires dependence on the point to be forgotten. To analyze this aspect, the unlearning point from step 1 is substituted with a randomly selected training point paired with its reversed label, resulting in the subsequent prompt: "$[Random\ Train\ Input]_0\ [Flipped\ Label]_0\ \backslash n\ [Input\ 1]_1\ [Label\ 1]_1\ \backslash n \cdots [Input\ s]_s$ $[Label\ s]_s\ [Query\ Input]_{s+1}$". We call this setting `Random ICUL(s)`.

These results are summarized in Figure 4, while additional evidence is provided in Appendix C.

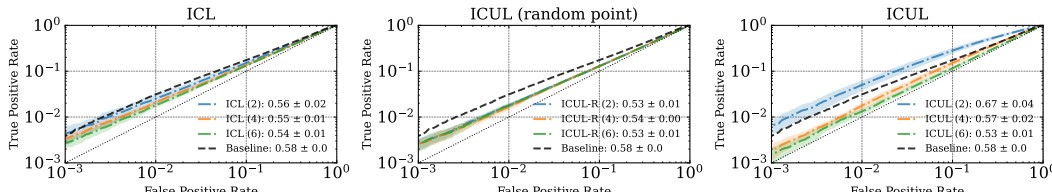

Figure 4: **Sensitivity analysis.** We plot MI attack performances as in Figure 6, this time across different context constructions described in Section 5.4 for the 1.1B Bloom model on the SST-2 dataset. The closer to the diagonal the better.

**ICL.** Here we empirically study the effect of label flipping on unlearning success. A comparison of the standard `ICL` approach (Figure 4, left), where the label of the point we aim to remove is kept unchanged, with our proposed `ICUL` method (Figure 4, right) illustrates that label flipping is a crucial factor that pushes the `ICUL` curve closer to the random guessing benchmark. This finding highlights the essential role of label flipping in successful unlearning and challenges recent studies that explore its significance in ICL (Min et al., 2022; Wei et al., 2023). While these studies propose that only large-scale language models can modify their predictions, our results suggest that smaller LLMs can adjust their predictions to mimic an output distribution that has never seen the removal point before.

**Varying context length.** We evaluate the effect of context length on the success of unlearning, as illustrated in the far-right plot of Figure 4. With shorter context lengths, such as 2, the reversed label of the forget point typically leaves an overly negative impact on the model's confidence scores. This generally results in poorer average performance than the `Baseline`, as shown by the comparison of their AUC scores (e.g., `ICUL(2)` scores at 0.67 while `Baseline` at 0.58). Furthermore, context lengths of this size are often not sufficient enough to reduce TPRs at FPR levels of $\{10^{-3}, 10^{-2}, 10^{-1}\}$ down to the level of random guessing benchmark. On the other hand, 4 or 6 additional context examples tend to yield the best performance. Further empirical evidence validating these observations across all datasets and model sizes is provided in Figure 7 of Appendix C.

**Dependence on forget point.** Finally, we examine whether the point intended for deletion needs to be part of the context. Evidence supporting this requirement is displayed by comparing the middle and right plots in Figure 4. This comparison highlights that in the low FPR regime at or below $10^{-2}$, our proposed `ICUL` method substantially surpasses the `ICUL` that uses a random point.

## 6 CONCLUSION

In this work, we suggested a novel class of unlearning problems that occurs in LLMs when there is no access to model parameters. Towards this paradigm, we propose a new unlearning method called In-Context UnLearning (`ICUL`). Our method effectively creates a model output distribution that mimics the scenario where a particular point was never part of the model's training dataset. To implement `ICUL`, we created prompts comprising the data point targeted for removal, its flipped label, as well as other accurately labeled instances. These prompts are then provided as inputs to the LLM during the inference stage. Our empirical results suggest that `ICUL` reliably removes the influence of training points on the model since an external auditor cannot reliably distinguish between held out points and training points that that should be removed from the model. Moreover, our empirical observations indicate that label flipping for in-context examples does have an impact on the model's output. This finding challenges earlier research that argued label flipping of context examples had an insignificant impact on smaller LLMs (Min et al., 2022; Wei et al., 2023). Future research will seek to extend our methodology to larger datasets and models, while also exploring the potential of

unlearning multiple points. Consequently, this work establishes a fundamentally novel perspective for the field of machine unlearning.

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

# A  REPRODUCIBILITY

We finetuned 80 models across two different model sizes and 4 different data sets; 40 models were finetuned using Bloom560M and another 40 models were finetuned using Bloom 1.1B. On average, finetuning took approximately 1 hour per model, which makes for 80 GPU hours. Regarding the main experiments, we conducted unlearning using both GA and ICUL. First, for ICUL we ran inference across 3 context length configurations across 80 models and each run took 2 hours on average. This amount to 480 GPU hours. Second, for GA, the situation was very similar. Updating the models using GA across 3 learning rate configurations for all 80 models where each run took approximately 2 hours amounts to another 480 GPU hours. Finally, we ran the additional sensitivity experimnts on SST-2 using Bloom 1.1B. These experiments were conducted for 10 models, using 3 context length configurations and 3 sensitivity setups, where each model run took approximately 1.5 hours, which makes for a total of 135 GPU hours. In total, we used 1175 GPU hours which approximately amounts to 49 GPU days. Note that these numbers include run times to find competitive learning rates and context lengths.

# B  ADDITIONAL RELATED WORKS

Here we briefly discuss additional related works that study (supervised) in-context learning theoretically (Xie et al., 2022; Akyürek et al., 2023; Von Oswald et al., 2023; Nagler, 2023; Zhang et al., 2023; Mahankali et al., 2023; Panigrahi et al., 2023; Ahn et al., 2023). Initially, Garg et al. (2022) empirically showed that linear transformers can learn simple function classes like linear regressors in-context. Inspired by these observations, Von Oswald et al. (2023) puts forth a weight construction for trained transformers that implements a single step of gradient descent in a forward pass, which has subsequently been studied in more detail showing that the corresponding weight construction is globally optimal (Zhang et al., 2023; Mahankali et al., 2023; Ahn et al., 2023) and that gradient flow reaches this optimum (Zhang et al., 2023).

# C  ADDITIONAL EXPERIMENTAL RESULTS

## C.1  SENSITIVITY OF ICUL TO LLM CHOICE AND SIZE

Here, we discuss how sensitive the ICUL results are to choice of underlying generative model. To this end, we consider Figures 5a and 5b which compare the unlearning efficacy across two models of similar size, a Bloom model with 1.1B parameters and a Pythia model with 1B parameters (Biderman et al., 2023). It becomes clear that ICUL works well regardless of model choice. Next, we demonstrate that ICUL works on larger SOTA LLMs such as Llama-2, which is the current best-in-class LLM when considering the size of 7B parameters (Touvron et al., 2023). These results are presented in Figure 5c. Again, the performance of ICUL traces close to the diagonal that represents a random guess probability of whether a point intended for removal is still part of the model.

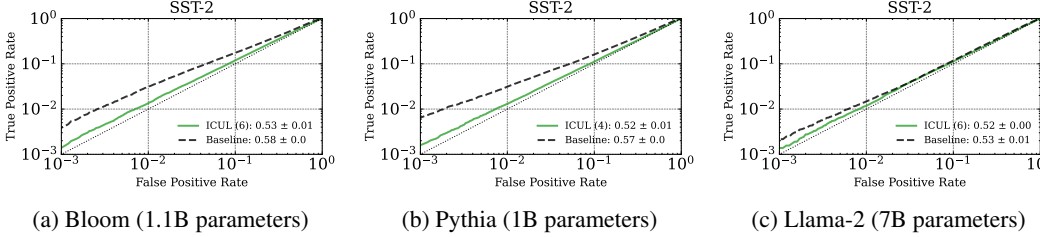

(a) Bloom (1.1B parameters)    (b) Pythia (1B parameters)    (c) Llama-2 (7B parameters)

Figure 5: **Comparing unlearning success of ICUL across different LLMs using log scaled ROC curves.** The closer to the diagonal the better, which amounts to the adversary randomly guessing whether a given point is (still) part of the model or not. For the green curves, the MI attacks were run against the updated models $\tilde{f}$, which were updated using ICUL (solid green). The black dashed line represents the baseline performance of not removing the point where the same attack is run on the model $f_{\theta(S)}$, as described in Section 5.1.

| Dataset | Method | Bloom 1.1B | | | Pythia 1B | | | Llama2 7B | | |
|---------|--------|-------|--------|------|-------|--------|------|-------|--------|------|
| | | Train | Forget | Test | Train | Forget | Test | Train | Forget | Test |
| SST-2 | ICUL | $0.925 \pm 0.018$ | $0.903 \pm 0.016$ | $0.881 \pm 0.015$ | $0.853 \pm 0.022$ | $0.862 \pm 0.025$ | $0.791 \pm 0.019$ | $0.914 \pm 0.065$ | $0.903 \pm 0.05$ | $0.882 \pm 0.058$ |
| | Baseline | $0.966 \pm 0.002$ | $0.966 \pm 0.002$ | $0.92 \pm 0.002$ | $0.958 \pm 0.018$ | $0.958 \pm 0.018$ | $0.874 \pm 0.015$ | $0.932 \pm 0.058$ | $0.932 \pm 0.058$ | $0.901 \pm 0.052$ |

Table 3: **Classification accuracy on train, forget and test points across different LLMs for the SST-2 dataset.** For the larger Llama2 7B model, the performance gap between predictions on test points using `ICUL` and those using `Baseline` is most narrow.

## C.2 Forgetting Multiple Points

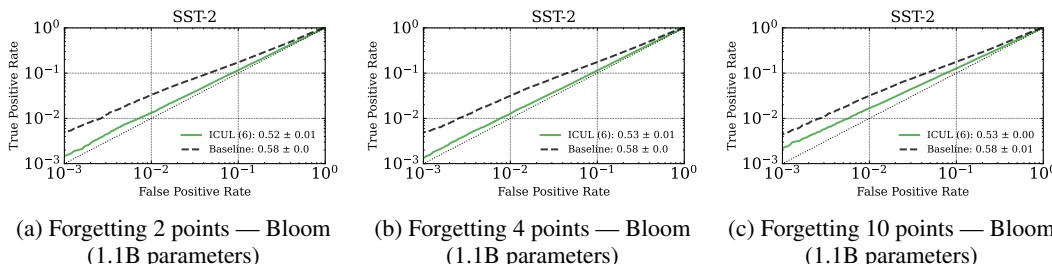

(a) Forgetting 2 points — Bloom (1.1B parameters)   (b) Forgetting 4 points — Bloom (1.1B parameters)   (c) Forgetting 10 points — Bloom (1.1B parameters)

Figure 6: **Comparing unlearning success of `ICUL` across different sizes of the forget set using log scaled ROC curves.** The closer to the diagonal the better, which amounts to the adversary randomly guessing whether a given point is (still) part of the model or not. For the green curves, the MI attacks were run against the updated models $\hat{f}$, which were updated using `ICUL` (solid green). The black dashed line represents the baseline performance of not removing the point where the same attack is run on the model $f_{\theta(S)}$, as described in Section 5.1. The numbers in brackets denote the best parameters and the numbers after that show the AUC $\pm 1$ standard deviation across 10 evaluation runs. Shades indicate $\pm 1$ standard deviation across 10 evaluation runs.

| Dataset | Method | 2 Deletions | | | 4 Deletions | | | 10 Deletions | | |
|---------|--------|-------|--------|------|-------|--------|------|-------|--------|------|
| | | Train | Forget | Test | Train | Forget | Test | Train | Forget | Test |
| SST-2 | ICUL | $0.903 \pm 0.041$ | $0.878 \pm 0.059$ | $0.861 \pm 0.036$ | $0.915 \pm 0.025$ | $0.894 \pm 0.03$ | $0.872 \pm 0.017$ | $0.926 \pm 0.014$ | $0.92 \pm 0.017$ | $0.882 \pm 0.012$ |
| | Baseline | $0.965 \pm 0.002$ | $0.965 \pm 0.002$ | $0.919 \pm 0.003$ | $0.965 \pm 0.001$ | $0.965 \pm 0.001$ | $0.92 \pm 0.003$ | $0.967 \pm 0.003$ | $0.967 \pm 0.003$ | $0.92 \pm 0.002$ |

Table 4: **Classification accuracy on train, forget and test points across different sizes of deletion requests when the LLM is Bloom with 1.1B parameters.**

## C.3 Optimizing Hyperparameters

Both `ICUL` and `GA` have each one crucial parameter, namely the context length and the learning rate. Here we investigate the impact that these parameters have on unlearning success and model performance.

**Vary the context length on `ICUL`.** For `ICUL`, changing the context length can significantly improve results in terms of unlearning success as seen in Figure 7. In terms of model performance, the situation is more nuanced as can be seen from Figure 8. While the context length has clear impact on forget points, test points seem to be impacted very little by the number of context examples.

**Vary the learning rate on `GA`.** For `GA`, changing the learning rate can dramatically improve results, where smaller learning rates usually significantly improve results in terms of unlearning success and model performance as shown in Figures 9 and 10.

## D Details on the Machine Unlearning Evaluation

Here we reproduce the discussion of Section 3.2 and provide additional details that have been left out of the discussion for brevity.

We define how we measure (approximate) unlearning. Our unlearning notion is that of (Ginart et al., 2019; Neel et al., 2021), but adapts the metric of membership inference attack success to

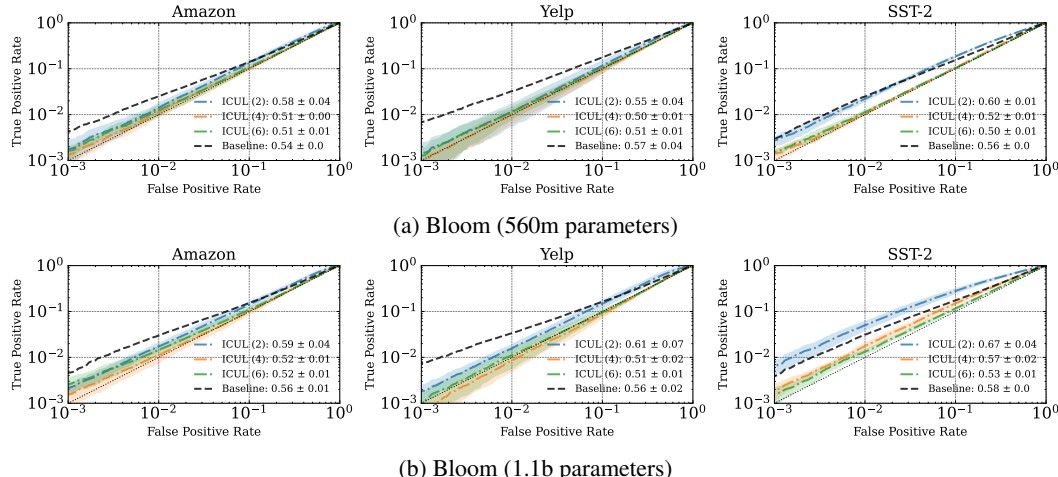

(a) Bloom (560m parameters)

(b) Bloom (1.1b parameters)

Figure 7: **Varying context length for ICUL**. Same setup as in Figure 6. We plot the MI attack performance using log scaled ROC curves across different datasets and model sizes. The MI attacks were run against the updated models $\bar{f}$, which was updated using ICUL. The closer to the diagonal, which amounts to the adversary randomly guessing whether a forget point is still part of the model or not, the better. The numbers in brackets denote the best parameters and the numbers after that show the AUC $\pm 1$ standard deviation across 10 evaluation runs. The black dashed line represents the baseline performance of not removing the point where the same attack is run on the model $f_{\theta(S)}$, as described in Section 5.1. Shades indicate $\pm 1$ standard deviation across 10 evaluation runs.

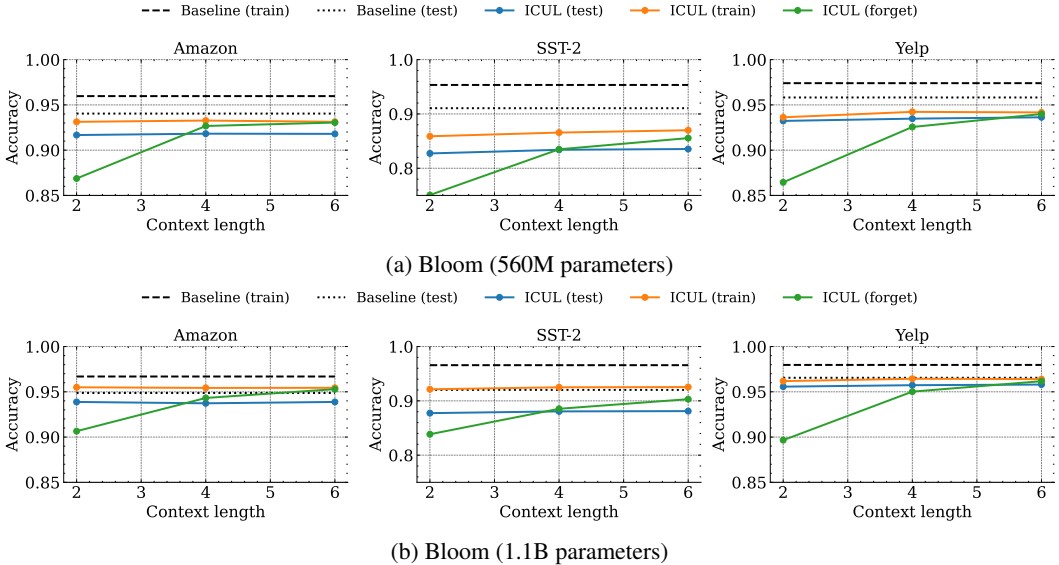

(a) Bloom (560M parameters)

(b) Bloom (1.1B parameters)

Figure 8: **Classification performance as we vary context length for ICUL**. We report classification accuracy on train, forget and test points across all data sets and model sizes. For better readability, $\pm 1$ standard deviation was excluded from the figure.

operationalize this definition (Goel et al., 2022; Golatkar et al., 2021). Let $S \subset \mathcal{S}^*$ denote the training set, sampled from a distribution $\mathcal{D} \in \Delta(\mathcal{S})$. Let $\mathcal{T} : \mathcal{S}^* \to \Theta$ be the (randomized) training algorithm that maps $S$ to a parameterized model $f_{\theta(S)}$. Further define the forget set as the subset of points to be forgotten from the trained machine learning model denoted by $S_f \subset S$. We define an unlearning procedure $\mathcal{U}$ that takes as input the model $f_{\theta(S)}$, the forget set $S_f$ of data samples that should be deleted, and the train set $S$ (and possibly some auxiliary information which we suppress), and outputs an updated model $\bar{f} \sim \mathcal{U}(f_{\theta(S)}, S, S_f)$. Denote the probability law of the training algorithm on input

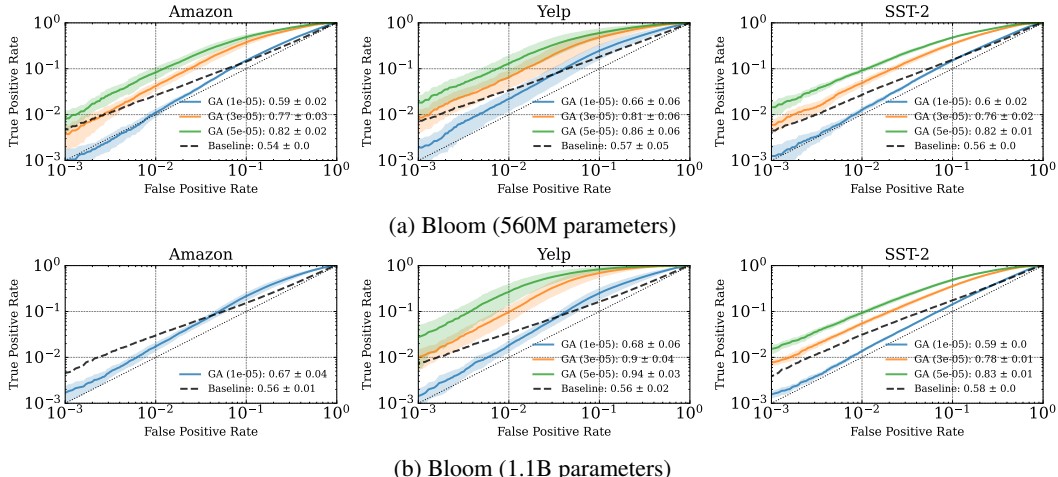

Figure 9: **Varying the learning rate for GA.** We plot the MI attack performance using log scaled ROC curves across different datasets and model sizes. The MI attacks were run against the updated models $\bar{f}$, which was updated using GA. The closer to the diagonal, which amounts to the adversary randomly guessing whether a forget point is still part of the model or not, the better. The numbers in brackets denote the best parameters and the numbers after that show the AUC $\pm 1$ standard deviation across 10 evaluation runs. The black dashed line represents the baseline performance of not removing the point where the same attack is run on the model $f_{\theta(S)}$, as described in Section 5.1. Shades indicate $\pm 1$ standard deviation across 10 evaluation runs.

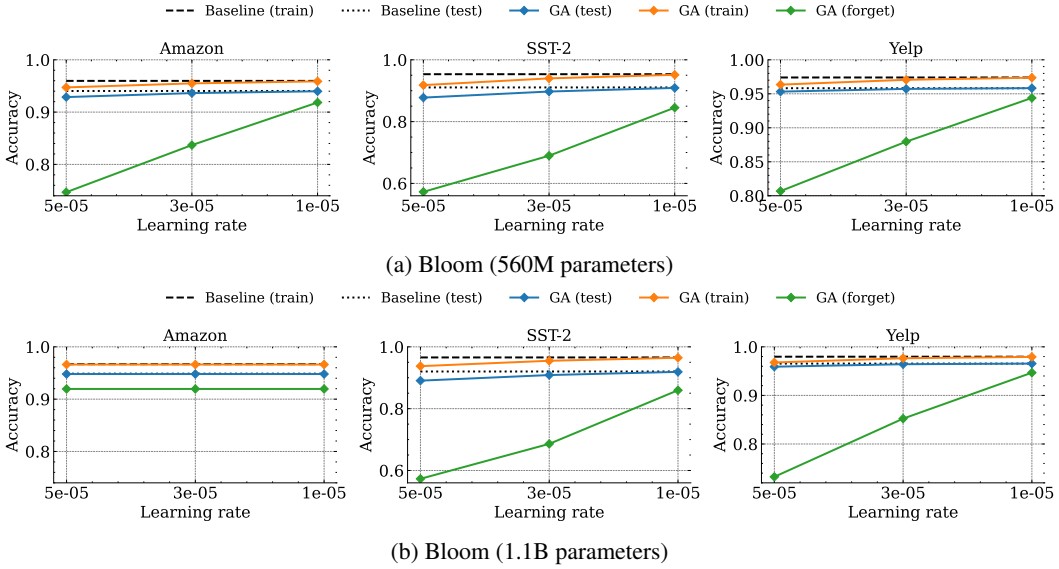

Figure 10: **Classification performance as we vary the learning rate for GA**. We report classification accuracy on train, forget and test points across all data sets and model sizes. For better readability, $\pm 1$ standard deviation was excluded from the figure.

$S$ by $p_S$, the law of the exact re-training algorithm by $p_{S \setminus S_f}$, and the law of the unlearning algorithm by $p_{\mathcal{U}}$. As first formalized in Ginart et al. (2019), the goal of an approximate unlearning algorithm is to produce $p_{\mathcal{U}} \approx p_{S \setminus S_f}$, or equivalently where $d(p_{S \setminus S_f}, p_{\mathcal{U}})$ is small for some distance measure between distributions $d$. Empirically verifying whether $d(p_{S \setminus S_f}, p_{\mathcal{U}})$ is small is difficult for two reasons: i) For computational reasons we do not have direct access to samples from $p_{S \setminus S_f}$, and ii) even if we did these distributions are extremely high dimensional and so we cannot compare them efficiently.

We address issue (i) by approximating the re-training distribution via sample-splitting (described in more detail in Appendix D); by training multiple models on splits of the data that do not contain $S_f$, we can approximate samples from $p_{S\setminus S_f}$. This approach is known as training "shadow-models" and has been employed for membership inference in Carlini et al. (2022); Shokri et al. (2017). We address (ii) by re-formulating the problem of bounding $d(p_\mathcal{U}, p_{S\setminus S_f})$ as a hypothesis testing problem. Le Cam's Lemma (see Theorem 2.2 in Tsybakov (2008)) establishes a correspondence between $d(p_\mathcal{U}, p_{S\setminus S_f})$ and the ability of an optimal hypothesis test to distinguish $p_\mathcal{U}$ from $p_{S\setminus S_f}$ based on a single sample. More specifically, we imagine a model $f$ is sampled from $p_\mathcal{U}$ with probability $1/2$ else from $p_\mathcal{U}$ with probability $1/2$, and conduct a hypothesis test to determine which distribution $f$ was sampled from:

$$\mathrm{H}_0 : f \sim p_{S\setminus S_f} \text{ vs. } \mathrm{H}_1 : f \sim p_\mathcal{U}. \tag{3}$$

Rejecting the null hypothesis corresponds to inferring that $f$ was not from the re-training distribution. The Neyman-Pearson lemma (Neyman & Pearson, 1933) asserts that the optimal hypothesis test at a predetermined false-positive rate involves thresholding the likelihood-ratio test $\Lambda$:

$$\Lambda = \frac{p_\mathcal{U}(f)}{p_{S\setminus S_f}(f)}. \tag{4}$$

As discussed, approximating Equation 4 is intractable due to the high dimensionality of $f$, and so we follow recent work on MIAs, that instead takes the likelihood ratio with respect to the distribution of losses on the forget points $S_f$ for both models. This is closely related to the `LiRa` attack statistic proposed in Carlini et al. (2022), but differs critically in that the numerator considers the model produced by training on $S_f$ *and then unlearning* via $\mathcal{U}$ rather than the model that results after training. When then define the `LiRA-Forget` statistic $\hat{\Lambda}$:

$$\hat{\Lambda} = \frac{\prod_{(\mathbf{x},)\in S_f} p_\mathcal{U}\big(\ell(f(\mathbf{x}),\mathbf{y})\big)}{\prod_{(\mathbf{x},)\in S_f} p_{S\setminus S_f}\big(\ell(f(\mathbf{x}),\mathbf{y})\big)}, \tag{5}$$

where $\ell$ denotes an appropriate loss function. As in these recent works we approximate the univariate distributions on losses in the numerator and denominator of (5) via sample-splitting. Specifically we fine-tune models on sub-sampled datasets that either contain or do not contain $S_f$. To approximate the numerator, on the datasets that do contain $S_f$, we run $\mathcal{U}$ to unlearn $S_f$, and then compute the updated model's loss on $S_f$. To approximate the denominator, we simple take the models that were not trained on $S_f$ and compute their losses on $S_f$. As in Carlini et al. (2022) we model the logit of the model's confidence as normal, and use these transformed confidences to estimate the likelihood ratio. Further details are provided in Appendix D.

We have described how to compute our unlearning success statistic $\hat{\Lambda}$, but it remains to discuss what values of $\hat{\Lambda}$ should be considered "successful". We continue our analogy to recent work in evaluating membership inference attacks, and follow the paradigm introduced in (Carlini et al., 2022) that focusing on true positive rates (in this case of predicting that the loss came from the unlearned model) at low false positive rates as the most intuitive measure of MIA attack success. In addition to plotting the full log-log ROC curves (Figures 3a and 3b) we also report the AUC. Unlike in the MIA context, where a successful attack has an AUC $\gg .5$, and an ROC curve that is above the diagonal even at very low FPRs, in our setting a successful unlearning algorithm corresponds to the failure of the LRT, and so we hope to see ROC curves that are very close to the diagonal even at low FPRs.

**Operationalizing the Likelihood-ratio Audit.** Operationalizing the likelihood ratio test from (5) requires access to the distribution of losses under the null and alternative hypotheses. While analytical solutions are usually not available, we can readily get large samples from these two distributions. In an ideal scenario, this entails that we would need to fit as many re-train models and unlearned models as possible for every forget set of interest. Since this approach becomes computationally too burdensome, we use the following two-step approximation:

**Approximating the distributions under $H_0$ and $H_1$.** Here we adapt the sample splitting procedure first introduced by Carlini et al. (2022) to forget sets with sizes $J$ greater than 1. We train $K$ shadow models on random samples from the data distribution $\mathcal{D}$ so that a fraction $p$ of these models are trained on the forget set $S_f = \{(\mathbf{x}_j, \mathbf{y}_j)\}_{j=1}^J$, and a fraction $(1-p)$ are not. In particular, we train shadow models on $K = 10$ subsets of $\mathcal{D}$ so that each forget set $S_f \in D$ appears in $K \cdot p$

subsets. This approach has the advantage that the same $K$ shadow models can be used to estimate the likelihood-ratio test for all the forget sets. Finally, we fit the parameters of two Gaussian distributions to the confidence scores of the retain models and the unlearned models on $S_f$. Across all experiments, we use $p = 0.5$ and $J = 1$.

**Model losses.** Instead of using the actual losses, we follow Carlini et al. (2022) and compute model confidences as $\phi(f(\mathbf{x}), \mathbf{y}) = \log(f(\mathbf{x})_y) - \log(\sum_{y'} f(\mathbf{x})_{y'})$ which the authors show yields the strongest empirical attack performance. This score compares the confidence the model assigns to the true class (e.g., 'positive') with the confidences the model assigns to all other classes (i.e., all other words from the approximately 250680 dimensional vocabulary). The higher the score is the more confident the model is in the correct prediction.

