# OpenReview forum: "In-Context Unlearning: Language Models as Few Shot Unlearners"
_ICLR.cc/2024/Conference — Submitted to ICLR 2024_

### Official Review · Reviewer_fKNk · 2023-10-30

**Soundness:** 3 good
**Presentation:** 2 fair
**Contribution:** 3 good
**Rating:** 5
**Confidence:** 4

**Summary:**

The authors introduce an innovative method termed 'Incontext machine unlearning.' A standout feature of this method is its ability to operate without needing access to the model's parameters, instead relying on in-context querying during inference. The crux of their work is a unique approach to unlearn specific training data points. Instead of resorting to computationally expensive retraining or finetuning, they suggest querying the language model with a carefully curated prompt during inference. This prompt is crafted to give the impression to the language model that it hasn't encountered a specific training instance. To implement this, upon receiving a deletion request, they recommend flipping the label of the targeted instance and appending it accordingly. Subsequent steps involve incorporating a set of randomly selected labeled example pairs, followed by the query input. The intention behind this structured query prompt is to effectively unlearn a designated training instance. While I find the core idea compelling, I feel the experiments section could benefit from enhanced clarity and depth. The presented results, especially the ROC curves, demonstrate the method's superiority over baseline techniques. The concept genuinely intrigues me, but I believe its presentation in the paper could be further refined for clarity and impact. I also think comparisons with prompt based adversarial attack methods would be necessary ([1] or related works). I have explained in detail about this below.

[1] Raman, Mrigank, et al. "Model-tuning Via Prompts Makes NLP Models Adversarially Robust." arXiv preprint arXiv:2303.07320(2023).

**Strengths:**

1) The idea of unlearning a subset of the training dataset through Incontext querying is particularly intriguing. This approach could significantly lessen the computational burden compared to previous machine unlearning techniques, like gradient ascent.

2) The presented ROC curves for the ICUL method are close to the diagonal, sufficiently backing the hypothesis that the proposed method is indeed successful in reducing the likelihood of the forget set belonging to the training set. (However, there are a few points that I would like to make regarding the presentation. Please see below.)

**Weaknesses:**

Firstly, thank you for an insightful work. I hope my comments will further strengthen your work.

1) Though, the presented idea is labeled as machine unlearning, I would like to point out that the presented approach queries the language model using a specific prompt designed to reduce the confidence of the model on a few training instances from the forget set. When queried using alternative prompts, there is no guarantee that the language model would perform with similar characteristics. Empirical evidence is the “Dependence on forget point” ablation study where using a random instance instead of the forget instance (i.e., a changed prompt) lead to inferior results. However, methods such as gradient ascent, though computationally more expensive than the proposed method, would lead to a reduction in likelihood of the forget set belonging to the training set irrespective of the query. In Fact they alter the weights such that the traces of forget set on the model weights would be minimized. Hence I believe this setting would ideally be more suitable for machine unlearning context. Speaking about the computational complexity of the gradient ascent method, I believe since the forget set cardinality is much less than the training set, the computational cost would not be very demanding and also it will be similar to finetuning language model on extremely low-resource scenarios.

2) I think it is necessary to compare with a few prompt based adversarial attack methods such as [1] or related methods. As stated above, the presented approach queries the language model using a specific prompt designed to reduce the confidence of the model on a few training instances from the forget set. So an alternative way to view the proposed method (deviating from machine unlearning) is to query a language model with a prompt such that the likelihood of the forget set belonging to the training set is reduced. Thus, it is like learning the perturbations to the input prompts that change the model predictions. Thus learn for such perturbations to the prompts only for the forget set. I believe that it serves a crucial baseline.

[1] Raman, Mrigank, et al. "Model-tuning Via Prompts Makes NLP Models Adversarially Robust." arXiv preprint arXiv:2303.07320(2023).

3) The presentation of the experiments section needs to be improved. For instance, i) Section 4.2 is redundant and is better explained in the ablations section. ii) In table 1, ICUL is mentioned without the query length (s) however it is mentioned with the query length in table 2. It is important to maintain consistency. Also it makes it more clear for the reader to mention what it means by ICUL (s) in the caption of the tables.

4) Baselines are not clearly mentioned in the experiments section. For instance, GA is not clearly mentioned. Also when mentioning the ‘Baseline’ in section 5.2, “consists of the decision not to unlearn the point from the model”. It is not clear what it means. I am assuming that the authors are referring to "Compare train vs. held out samples on the initial model fθ(S)" in section 5.1. Maybe mentioning it directly where they first introduce will make it less confusing.

5) The experiments section requires further elaboration. While the authors assert that their method outperforms the baselines, they should delve into why this is the case. For example, the claim that their method consistently surpasses the GA baseline is supported by empirical evidence however remains unexplained. Intuitively, the GA method, which uses gradient ascent to intentionally forget the 'forget set', should be on par with or even superior to the proposed method. Clarifying this would provide valuable insight.

**Questions:**

1) The format for the incontext unlearning is as follows: “[Forget Input] [Flipped Label] \n [Input 1]1 [Label 1]1 \n · · · [Input s]s [Label s]s [Query Input]s+1 ”. My question is if I want to query on [Forget Input] again the format is as follows, “[Forget Input] [Flipped Label] \n [Input 1]1 [Label 1]1 \n · · · [Input s]s [Label s]s [Forget Input] ”. Is my understanding correct? If so, it would be interesting to know how many times the output of the language model is [Flipped Label]? We can understand the broader impact of such a querying by further knowing these dynamics.
2) Also from the results, it is evident that the method performs at par with the GA method however it is not really clear why the GA method performs worse than the Baseline (especially the sudden spike towards 10^(-1) FPR for Amazon and Yelp). Could you explain why that is the case? We are optimizing by performing gradient ascent so it has to be below baseline at least?
3) In section 4, it is mentioned that “For finetuning, we are using the standard causal language loss which encourages the model to predict the next token correctly”. So when the models are fine-tuned on the downstream dataset, why is the result on the train set low, especially for ICUL(6) on SST-2 dataset (Table-2).
4) Please correct the typos and ensure consistency in notation. For example, 'membership inference' is abbreviated in some instances, while in others, it's written out in full.
5) The authors claim "This finding challenges earlier research that argued label flipping of context examples had an insignificant impact on smaller LLMs " in the conclusions. It would be more insightful if the authors can point out why is the case? Or if there is any assumption difference between these works and the proposed work.

---

> ### Comment · Reviewer_fKNk · 2023-11-21
>
> Firstly, the method is heavily inspired from the paper "Larger language models do in-context learning differently". The authors propose that it works even for smaller models which challenges the observations of the earlier paper. However, it is not clear why it is the case. Were there any assumptions that were made in this paper that differed from the settings of the former paper?
>
> Though this paper is proposed as machine unlearning, I am not really convinced if it falls into the realms of machine unlearning since the model weights still preserve the information of the user. So information is never really lost. As I mentioned above, by changing the prompt you can always retrieve the information.
>
> Since my questions are not adequately answered, I am reducing the score from 6->5.

---

> > ### Author Response · Authors · 2023-11-23
> > **Response to Comment by Reviewer fKNk**
> >
> > We would like to thank the reviewer for actively engaging in the discussion and apologize for our late rebuttal response which was due to the key experiments taking several gpu days to run.
> >
> > **Clarification on Inspirations and Methodological Differences**
> >
> > > Firstly, the method is heavily inspired from the paper "Larger language models do in-context learning differently". The authors propose that it works even for smaller models which challenges the observations of the earlier paper. However, it is not clear why it is the case. Were there any assumptions that were made in this paper that differed from the settings of the former paper?
> >
> > Inspiration from [2] Clarified:
> > - We appreciate the reviewer's observation regarding the resemblance to [2]. However, it's crucial to clarify that our work fundamentally differs from [2]. While [2] explores the impact of random label flipping on a model's ability to make correct predictions, our focus is on unlearning through label flipping and subsequent evaluations using sample splitting and Membership Inference Attacks (MI Attacks). The distinction lies in our methodology, aiming to answer specific questions about unlearning capabilities rather than studying the impact of label flipping on classification accuracy.
> >
> > Different Research Objectives and Evaluations:
> > - The divergence in research objectives is key. [2] employs random label flipping as an evaluation technique, whereas we propose label flipping as a targeted unlearning procedure. Our approach enables us to make explicit claims about ICUL's unlearning capabilities. The varying focuses emphasize that their evaluation does not directly facilitate claims about unlearning success. Our methodology, with unlearning through label flipping and subsequent evaluations, contributes uniquely to the understanding of achieving unlearning in-context.
> >
> > Nuanced Findings Regarding Model Size:
> > -  In contrast to the findings in [2], our research suggests that the performance of large language models (LLMs) may not be solely determined by their size. While [2] highlighted the tendency of smaller LLMs to neglect prompt context, emphasizing the superior context utilization of larger models for correct model classifications, our work introduces a nuanced perspective wrt. to unlearning questions. We argue that the capacity to selectively forget specific instances from the training set is not exclusive to larger models, thus our work complements the picture by providing evidence that not only increased model size allows for effective knowledge overwriting.
> >
> > **Real-World Scenario Clarification**
> >
> > >Though this paper is proposed as machine unlearning, I am not really convinced if it falls into the realms of machine unlearning since the model weights still preserve the information of the user. So information is never really lost. As I mentioned above, by changing the prompt you can always retrieve the information.
> >
> > - Recall that our methodology is tailored to the real-world constraints of 'ML as a Service' platforms. There, the model owner always prepends the required unlearning contexts in front of the queries since it is the premise that the LLM operates as black-box (see Figure 1). In such scenarios, where model parameters are inaccessible, and standard unlearning techniques relying on gradient ascent are infeasible, our approach of pre-appending required unlearning contexts becomes crucial. For example, a bank finetunes a GPT3.5 model through OpenAI’s API for the task of deciding credit worthiness of customers. The bank would need to finetune the entire model from scratch when deletion requests come in since standard techniques like gradient ascent do require parameter access which we do not have when using the API. Hence, our technique is particularly useful when standard unlearning methods that operate via gradient descent on the model’s parameters cannot be implemented as the lack of parameter access impedes traditional unlearning methods.
> >
> > We hope this response has sufficiently addressed the concerns raised. Given the clarifications provided, we kindly request the reviewer to reconsider their score in light of the distinct contributions and considerations outlined in our response.
> >
> > ---
> > **References**
> >
> > [1] Joel Jang, Dongkeun Yoon, Sohee Yang, Sungmin Cha, Moontae Lee, Lajanugen Logeswaran, and Minjoon Seo. Knowledge unlearning for mitigating privacy risks in language models. In Proceedings of the 61st Annual Meeting of the Association for Computational Linguistics (ACL), 2023
> >
> > [2] Jerry Wei, Jason Wei, Yi Tay, Dustin Tran, Albert Webson, Yifeng Lu, Xinyun Chen, Hanxiao Liu, Da Huang, Denny Zhou, et al. Larger language models do in-context learning differently. arXiv:2303.03846, 2023

---

> ### Author Response · Authors · 2023-11-22
> **Response to comment by Reviewer fKNk**
>
> We are working on additional experiments and responses to address reviewer questions. We will be posting them very soon. Thanks for your patience.

---

> ### Author Response · Authors · 2023-11-22
> **Response to Reviewer fKNk (Part 1)**
>
> We are very grateful for the reviewer's positive and thoughful comments and believe that they have contributed to improving our work. Below we address individual points raised by the reviewer.
>
> **Querying with alternative prompts**
>
> > When queried using alternative prompts, there is no guarantee that the language model would perform with similar characteristics.
>
> This is correct. However, recall that our methodology is tailored to the real-world constraints of 'ML as a Service' platforms. There, the model owner always prepends the required unlearning contexts in front of the queries since it is the premise that the LLM operates as black-box (see Figure 1). In such scenarios, where model parameters are inaccessible, and standard unlearning techniques relying on gradient ascent are infeasible, our approach of pre-appending required unlearning contexts becomes crucial. For example, a bank finetunes a GPT3.5 model through OpenAI’s API for the task of deciding credit worthiness of customers. The bank would need to finetune the entire model from scratch when deletion requests come in since standard techniques like gradient ascent do require parameter access which we do not have when using the API. Hence, our technique is particularly useful when standard unlearning methods that operate via gradient descent on the model’s parameters cannot be implemented as the lack of parameter access impedes traditional unlearning methods.
>
>
> **Computational aspects of unlearning**
> > Speaking about the computational complexity of the gradient ascent method, I believe since the forget set cardinality is much less than the training set, the computational cost would not be very demanding and also it will be similar to finetuning language model on extremely low-resource scenarios.
>
> Contrary to the hypothesis regarding the computational aspects of unlearning, our experiments demonstrate the distinct memory requirements for gradient ascent (GA) and our proposed In-Context Unlearning (ICUL). In response to the reviewers hypothesis, we consider unlearning of sets of size 1 on the Llama2 7b model fine-tuned on the SST2 dataset (see Appendix C.1). Here, ICUL proves to be resource-efficient. Running ICUL on a single Tesla V100 GPU with 32GB of RAM contrasts sharply with the demands of gradient ascent (GA), which necessitates an A100 GPU with 80GB of RAM.
>
> **Prompt based adversarial attacks**
>
> > I think it is necessary to compare with a few prompt based adversarial attack methods such as [1] or related methods.
>
> We appreciate the reviewer's insightful suggestion regarding alternative ICUL prompts. Indeed, our approach can be seen as a perturbation to the query point that reduces the likelihood of the query point belonging to the training set. We acknowledge that there may be other effective unlearning contexts. Unfortunately, due to resource constraints, we were unable to run the proposed method from [1] in addition to our new experiments (see Appendix C) as an alternative ICUL prompt. However, we thank the reviewer for this valuable suggestion and are committed to exploring adversarially perturbed prompts, as proposed in [1], in our future work. This aligns with our acknowledgment that there could be multiple routes to achieve unlearning, and we look forward to further investigations in this direction.
>
> **Presentation**
> > The presentation of the experiments section needs to be improved.
>
> Thank you for this suggestion. We have updated the manuscript by moving section 4.2 to Section 5.4 where we conduct the sensitivity analysis. We also added three corresponding values for s that were used for ICUL in Table 1, one value of s for each dataset.
>
>
> **Correcting typos**
> > Please correct the typos and ensure consistency in notation. For example, 'membership inference' is abbreviated in some instances, while in others, it's written out in full.
>
> We made sure that MI is used consistently throughout. Please let us know if the reviewer has found additional typos.
>
> **Mentioning baselines**
> > Baselines are not clearly mentioned in the experiments section. For instance, GA is not clearly mentioned [...]
>
> We have made this more clear now. We have highlighted the abbreviation gradient ascent (GA) in the preamble of  section 5 again. We also introduce the term Baseline in Section 5.1 more clearly. Please refer to the text highlighted in green for the concrete section we made to Section 5.

---

> ### Author Response · Authors · 2023-11-22
> **Response to Reviewer fKNk (Part 2)**
>
> **Further elaborations**
> > The experiments section requires further elaboration. While the authors assert that their method outperforms the baselines, they should delve into why this is the case.
> > Also from the results, it is evident that the method performs at par with the GA method however it is not really clear why the GA method performs worse than the Baseline (especially the sudden spike towards 10^(-1) FPR for Amazon and Yelp).
>
> When using GA, we have been strictly following the methodology proposed in [1]. We performed gradient ascent for one epoch using adamw on the forget set $S_f$ tuning learning rates in the range of [5e-05, 3e-05, 1e-05] as a hyperparameter, consistent with the approach in [1]. It is conceivable that using different optimizers would avoid the bump and could lead to improved unlearning results for GA.
>
> **Questions**
>
> > My question is if I want to query on [Forget Input] again the format is as follows, “[Forget Input] [Flipped Label] \n [Input 1]1 [Label 1]1 \n · · · [Input s]s [Label s]s [Forget Input] ”. Is my understanding correct? If so, it would be interesting to know how many times the output of the language model is [Flipped Label]? We can understand the broader impact of such a querying by further knowing these dynamics.
>
> Yes, your understanding is correct! Also, the output flips in 1 to 5 percent of all cases. While the output is usually not flipped, it is the confidence in the true class that decreases.
>
> > In section 4, it is mentioned that “For finetuning, we are using the standard causal language loss which encourages the model to predict the next token correctly”. So when the models are fine-tuned on the downstream dataset, why is the result on the train set low, especially for ICUL(6) on SST-2 dataset (Table-2).
>
> This phenomenon is most pronounced on the smaller Bloom 560M model (3rd column in Table 2). The larger Bloom 1.1B model manages to use the correctly labeled examples from its context more effectively than the smaller 560M model (6th column in Table 2).
>
> > The authors claim "This finding challenges earlier research that argued label flipping of context examples had an insignificant impact on smaller LLMs " in the conclusions. It would be more insightful if the authors can point out why is the case? Or if there is any assumption difference between these works and the proposed work.
>
> In contrast to the findings in [2], our research suggests that the performance of large language models (LLMs) may not be solely determined by their size. While [2] highlighted the tendency of smaller LLMs to neglect prompt context, emphasizing the superior context utilization of larger models, our work introduces a nuanced perspective. We argue that the capacity to selectively forget specific instances from the training set is not exclusive to larger models, thus challenging the notion that only increased model size allows for effective knowledge overwriting.
>
> Further, these two works fundamentally differ in the questions they aim to answer. While both our work and the the work from [2] utilize label flipping, it's crucial to emphasize the divergence in our research objectives and evaluations. The commonality lies in the application of label flipping, but the motivations behind its use and the questions addressed are markedly different. In [2], the authors employ *random label flipping* primarily as an evaluation technique to study the impact of label flipping itself on model predictions. In contrast, we propose *label flipping of specific points targeted for removal* as an unlearning procedure and subsequently employ a combination of sample splitting and Membership Inference Attacks (MI Attacks) for our evaluation. This nuanced approach allows us to make specific claims about the unlearning capabilities of ICUL. In summary, their focus on the impact of label flipping do not facilitate claims about unlearning success. Our methodology, centered around unlearning through label flipping and the subsequent evaluations, provides a distinct angle that contributes to the broader understanding of how unlearning can be achieved in-context.
>
> ----
>
> **References**
>
> [1] Joel Jang, Dongkeun Yoon, Sohee Yang, Sungmin Cha, Moontae Lee, Lajanugen Logeswaran, and Minjoon Seo. Knowledge unlearning for mitigating privacy risks in language models. In Proceedings of the 61st Annual Meeting of the Association for Computational Linguistics (ACL), 2023
>
> [2] Jerry Wei, Jason Wei, Yi Tay, Dustin Tran, Albert Webson, Yifeng Lu, Xinyun Chen, Hanxiao Liu, Da Huang, Denny Zhou, et al. Larger language models do in-context learning differently. arXiv:2303.03846, 2023

---

### Official Review · Reviewer_Awru · 2023-10-31

**Soundness:** 3 good
**Presentation:** 3 good
**Contribution:** 2 fair
**Rating:** 5
**Confidence:** 3

**Summary:**

The paper introduces an innovative method called "In-Context Unlearning" (ICUL) designed for large language models (LLMs). ICUL facilitates the unlearning of specific training instances by providing contextual inputs at the inference stage, thereby eliminating the need for modifying model parameters. Experiments confirm that ICUL can effectively remove targeted training information while maintaining performance levels.

**Strengths:**

* The paper brings a novel approach to the domain of machine unlearning, effectively leveraging the In-Context Learning (ICL) paradigm to achieve unlearning in text classification tasks without any modification to model parameters.
* The design of ICUL requires minimal computational resources, offering a cost-effective alternative to traditional unlearning methods that involve model retraining.
* The paper includes a thorough ablation study, which affirms the effectiveness of the key components in ICUL, specifically label flipping and the addition of unforgotten samples.

**Weaknesses:**

* The experiments are conducted only on Bloom's 560M and 1.1B models, lacking comparison with newer, larger models like LLaMA, Falcon, or ChatGPT. Moreover, ICUL is compared to only a single benchmark method (GA), which in some settings even outperforms ICUL, making it difficult to assert ICUL's superiority.
* The method is tailored for binary classification tasks and does not easily extend to multi-class or more complex NLP tasks. The paper also falls short in clarifying the real-world scenarios and problems it aims to address, leaving its practical utility ambiguous.
* The method's design, focusing on unlearning single samples in classification tasks, makes it less suitable for handling large volumes of unlearning requests in realistic scenarios.
* The paper doesn't delve into the theoretical underpinnings that explain why ICUL is effective at unlearning, leaving a gap in our understanding of the method's robustness.
The paper fails to compare or reference highly related work in the area of concept erasure in NLP, e.g., 1,2,3.

1. Shauli Ravfogel, Francisco Vargas, Yoav Goldberg, Ryan Cotterell. Adversarial Concept Erasure in Kernel Space. EMNLP 2022
2. Shauli Ravfogel, Michael Twiton, Yoav Goldberg, Ryan Cotterell. Linear Adversarial Concept Erasure. ICML 2022
3. Nora Belrose, David Schneider-Joseph, Shauli Ravfogel, Ryan Cotterell, Edward Raff, Stella Biderman. LEACE: Perfect linear concept erasure in closed form.

**Questions:**

How is ICUL designed to scale for real-world applications that demand continuous unlearning?

---

> ### Author Response · Authors · 2023-11-22
> **Response to Reviewer Awru (Part 1)**
>
> We are grateful for the reviewer's critical and thoughful comments and believe that they have substantially contributed to improving our work. Below we address individual points raised by the reviewer.
>
> **Single benchmark**
>
> > [...] ICUL is compared to only a single benchmark method (GA) [...]
>
> 1) *Current SOTA unlearning methods rely on the Hessian and do not apply to LLMs*: Unlearning training points in Language Models (LLMs) is an emerging area, and the limited availability of viable alternatives, exemplified by [2], underscores the current landscape. The primary reason for this scarcity is rooted in the design constraints of existing methods. Many prevalent approaches, both exact and approximate, necessitate the computation and inversion of the Hessian matrix ([6,7]). However, for LLMs, the computation of the Hessian and its inverse become computationally infeasible. Notably, the alternative used by [2], as suggested in [8], leverages gradient ascent on the points targeted for deletion, avoiding the need for Hessian matrix computations.
>
> 2) *ICUL is close to the optimal performance*: In terms of unlearning efficacy, it's crucial to highlight that our method achieves performance comparable or extremely close to the optimal scenario of randomly guessing whether the forgotten point is still part of the training set (refer to Figure 3).Furthermore, concerning overall model performance, both Gradient Ascent (GA) and ICUL exhibit similar performance for the larger Bloom 1.1B LLM. Both methods showcase classification accuracy that is remarkably close to the Baseline, where no unlearning has occurred. This parity underscores the effectiveness of ICUL in preserving model performance despite unlearning.
>
> 3) *No points of comparison*: Furthermore, our suggested approach is fundamentally different from any other unlearning mechanism in literature. As opposed to [2] and the additional works that the reviewer suggested for comparison [3-5], our work does not require updating of model parameters. Further, and more fundamentally, [3-5] focus on unlearning (human defined) concepts, while our work focuses on unlearning individual points, and hence [3-5] are not applicable in our work.
>
>
> **Comparison with newer and larger language models**
>
> > The experiments are conducted only on Bloom's 560M and 1.1B models, lacking comparison with newer, larger models like LLaMA, Falcon, or ChatGPT [...]
>
> Addressing the suggestion from the reviewer and to underscore the general applicability of our method across various large language models, we conducted additional experiments to evaluate the performance of our unlearning algorithm in forgetting points from the SST2 dataset additionally using Pythia 1B and LLAMA2 7B. For these experiments, we employed the same methodology for forgetting points as previously described in Section 4 of our paper. These additional results are depicted in Appendix C1 and show that ICUL successfully forgets points across different language model architectures. The results for GA are still running and will be included in the plots as soon as they are complete.
>
> **Forgetting multiple points**
>
> >The method's design, focusing on unlearning single samples in classification tasks, makes it less suitable for handling large volumes of unlearning requests in realistic scenarios.
>
> Our method is suitable for larger volumes of unlearning requests. Addressing the suggestion from the reviewer and to underscore the general applicability of our method across multiple deletion requests, we conducted additional experiments to evaluate the performance of our unlearning algorithm in forgetting  2, 4  and 10 points from the SST2 dataset using the 1.1B Bloom model. We employed the same methodology for forgetting multiple points as previously described in Section 4 of our paper. Specifically, we construct prompts by first flipping the labels on the designated points for forgetting (i.e., 2, 4 or 10), followed by the inclusion of additional correctly labeled examples. The results presented in Appendix C.2 of the updated manuscript confirm that ICUL maintains forgetting efficacy in this extended scenario. We are committed to add experimental results on all remaining models (Bloom 560M, Bloom 1.1B, Pythia 1B and Llama2 7B) and datasets (Amazon and Yelp) once they will finish running.

---

> > ### Author Response · Authors · 2023-11-22
> > **Response to Reviewer Awru (Part 2)**
> >
> > **Comparing ICUL and GA**
> >
> > > which in some settings even outperforms ICUL, making it difficult to assert ICUL's superiority
> >
> > Please note that ICUL is close to the optimal performance when for larger lanuage models like Bloom 1.1B. In terms of unlearning efficacy, it's crucial to highlight that our method achieves performance comparable or extremely close to the optimal scenario of randomly guessing whether the forgotten point is still part of the training set (refer to Figure 3).Furthermore, concerning overall model performance, both Gradient Ascent (GA) and ICUL exhibit similar performance for the larger Bloom 1.1B LLM. Both methods showcase classification accuracy that is remarkably close to the performance of the optimal benchmark, where no unlearning has occurred. This parity underscores the effectiveness of ICUL in preserving model performance despite unlearning.
> >
> > We would also like to highlight the fundamentally different memory requirements for GA and ICUL, especially if the LLMs become larger. For example, on Llama2 7b, since ICUL works by providing appropriate inputs at test time, it could be run using one Tesla V100 GPU with 32GB of RAM, while running GA requires access to a A100 GPU with 80GB of RAM.
> >
> >
> > **Theoretical results**
> >
> > >The paper doesn't delve into the theoretical underpinnings [...]
> >
> > Theoretical results for unlearning in Language Models (LLMs) pose a great challenge due to the nonconvex nature of these models. Obtaining theoretical insights becomes inherently complex in such scenarios. Consequently, we meticulously design the unlearning experiment outlined in Section 3.2. The primary goal is to scrutinize whether our proposed approach results in an output distribution akin to what one would observe if the model had never been exposed to the specific data points targeted for unlearning. This experimental setup allows us to pragmatically explore the outcomes of ICUL in the absence of tractable theoretical results.
> >
> > > [...] that explain why ICUL is effective at unlearning, leaving a gap in our understanding of the method's robustness,
> >
> > Although our work is empirical, we have a good understanding of how our method works. Through label flipping on the points targeted for unlearning, ICUL diminish the model's confidence specifically on these instances, aligning them more closely with the model's confidence as if the points were never part of the training set.
> >
> > **Updated related work**
> >
> > > The paper fails to compare or reference highly related work in the area of concept erasure in NLP, e.g., [[3-5]]
> >
> > Thanks for suggesting [3-5] as related work. The manuscript has been revised accordingly, with [3-5] now appropriately cited within the related work section (see additions in Section 2 which have been highlighted in green). There are several differences between [3-5] and our work. First, [3-5] focus on unlearning concepts like ‘gender’ or ‘color’ from a trained LLM which is different from our approach which unlearns individual points. Note that the approach we compare to [2] in our work also unlearns individual points. Further, [3-5] are fundamentally different in the sense that they all require whitebox model access, which our approach does not require.

---

> ### Author Response · Authors · 2023-11-22
> **Response to Reviewer Awru (Part 3)**
>
> **Practicality**
> > The paper also falls short in clarifying the real-world scenarios and problems it aims to address, leaving its practical utility ambiguous.
>
> Our methodology is tailored to the real-world constraints of 'ML as a Service' platforms. There, the model owner always prepends the required unlearning contexts in front of the queries since it is the premise that the LLM operates as black-box (see Figure 1). In such scenarios, where model parameters are inaccessible, and standard unlearning techniques relying on gradient ascent are infeasible, our approach of pre-appending required unlearning contexts becomes crucial. For example, a bank finetunes a GPT3.5 model through OpenAI’s API for the task of deciding credit worthiness of customers. The bank would need to finetune the entire model from scratch when deletion requests come in since standard techniques like gradient ascent do require parameter access which we do not have when using the API. Hence, our technique is particularly useful when standard unlearning methods that operate via gradient descent on the model’s parameters cannot be implemented as the lack of parameter access impedes traditional unlearning methods.
>
> > The method is tailored for binary classification tasks and does not easily extend to multi-class or more complex NLP tasks.
>
> Thanks for this suggestion. In this work, we have outlines the fundamental mechanism to unlearn in-context using binary classification problem. We are commited to extending our method beyond the binary case in follow-up work.
>
>
> We thank the reviewer again for their thoughtful comments and feedback. We hope we addressed all your questions/concerns/comments adequately. In light of our clarifications, please consider increasing your score.
>
> ---
> **References**
>
> [1] Biderman et al (2023, “Pythia: A Suite for Analyzing Large Language Models Across Training and Scaling”, Proceedings of the 40 th International Conference on Machine Learning (ICML), 2023
>
> [2] Joel Jang, Dongkeun Yoon, Sohee Yang, Sungmin Cha, Moontae Lee, Lajanugen Logeswaran, and Minjoon Seo. Knowledge unlearning for mitigating privacy risks in language models. In Proceedings of the 61st Annual Meeting of the Association for Computational Linguistics (ACL), 2023
>
> [3] Shauli Ravfogel, Francisco Vargas, Yoav Goldberg, Ryan Cotterell. Adversarial Concept Erasure in Kernel Space. EMNLP 2022
>
> [4] Shauli Ravfogel, Michael Twiton, Yoav Goldberg, Ryan Cotterell. Linear Adversarial Concept Erasure. ICML 2022
>
> [5] Nora Belrose, David Schneider-Joseph, Shauli Ravfogel, Ryan Cotterell, Edward Raff, Stella Biderman. LEACE: Perfect linear concept erasure in closed form.
>
> [6] Aditya Golatkar, Alessandro Achille, and Stefano Soatto. Forgetting outside the box: Scrubbing deep networks of information accessible from input-output observations. arXiv:2003.02960, 2020b.
>
> [7] Ayush Sekhari, Jayadev Acharya, Gautam Kamath, and Ananda Theertha Suresh. Remember what you want to forget: Algorithms for machine unlearning. In Advances in Neural Information Processing Systems, 2021
>
> [8] Seth Neel, Aaron Roth, and Saeed Sharifi-Malvajerdi. Descent-to-delete: Gradient-based methods for machine unlearning. In Proceedings of the 32nd International Conference on Algorithmic Learning Theory (ALT), 2021.

---

### Official Review · Reviewer_poKX · 2023-11-01

**Soundness:** 3 good
**Presentation:** 3 good
**Contribution:** 3 good
**Rating:** 6
**Confidence:** 2

**Summary:**

This paper looks at the problem of machine unlearning in the context of LLM. In particular, the authors look at in-context unlearning, where the forget example is fed into the model context window with a flipped sign, and no gradient update is necessary. The authors show that the proposed method is competitive regarding unlearning effectiveness and accuracy on the retain set.

**Strengths:**

The paper poses a very interesting problem and it can motivate future works. The paper is written nicely and it is easy to follow.

**Weaknesses:**

I found the particular setting in this paper to be a little restrictive (but it's not crucial since this paper is almost initiating a new setting) that only one example is forgetting.

**Questions:**

1. In eq(2), what is $\hat \theta$, does this mean in different scenarios, you test with different estimators, e.g. the ERM $\theta(S)$ and the unlearned model $\bar f$?
2. When you actually perform the LRT in eq(2), do you take each (x,y) as one test samples?
3. When you fine-tune with GA, what are the stopping criteria? Do you have a suggestion on some rule-of-thumb?
4. One related prior work should be included and discussed, see [1].

[1] KGA: A General Machine Unlearning Framework Based on Knowledge Gap Alignment

---

> ### Author Response · Authors · 2023-11-22
> **Response to Reviewer poKX**
>
> We are very grateful for the positive comments from the reviewer, which highlight the novelty of our suggested unlearning method. Below we address individual points raised by the reviewer.
>
> **Forgetting multiple points**
>
> >I found the particular setting in this paper to be a little restrictive (but it's not crucial since this paper is almost initiating a new setting) that only one example is forgetting.
>
> Addressing the suggestion from the reviewer and to underscore the general applicability of our method, we conducted additional experiments to evaluate the performance of our unlearning algorithm in forgetting 2, 4 and 10 points from the SST2 dataset using the 1.1B Bloom LLM. We employed the same methodology for forgetting multiple points as previously described in Section 4 of our paper. Specifically, we construct prompts by first flipping the labels on the designated points for forgetting (i.e., 2, 4 or 10), followed by the inclusion of additional correctly labeled examples. The results presented in Appendix C.2 of the updated manuscript confirm that ICUL maintains its strong forgetting efficacy in this extended scenario. We are committed to add experimental results on all remaining models (Bloom 560M, Bloom 1.1B, Pythia  1B and Llama2 7B) and datasets (Amazon and Yelp) once all these experimental runs are complete.
>
>
> **Approximating the likelihood ratio in equation  (2)**
>
> >In eq(2), what is $\hat{\theta}$, does this mean in different scenarios, you test with different estimators, e.g. the ERM and the unlearned model ?
>
> $\hat{\theta}$: This was a typo and should have been replaced by $\bar{\theta}$, the unlearned model as the nominrator represents the distribution of model losses after unlearning. To be more specifc, to approximate the univariate distributions in the numerator and denominator of equation (2), we employ a sample-splitting approach. Specifically, we fine-tune models on sub-sampled datasets that either include or exclude $S_f$. For the numerator, we apply $\mathcal{U}$ to unlearn $S_f$ on datasets containing it, followed by computing the updated model’s loss on $S_f$. For the denominator, we compute the losses on $S_f$ for models not trained on $S_f$. We have made this more clear by slightly rewriting Section 3.2.
>
>
> >When you actually perform the LRT in eq(2), do you take each (x,y) as one test samples?
>
> Given our use of shadow models as outlined above, to approximate the likelihood ratio, each (x,y) is anticipated to serve as a test sample in half of the shadow models and as a training sample in the remaining half.
>
>
> **Gradient Ascent on the Forget set**
>
> > When you fine-tune with GA, what are the stopping criteria? Do you have a suggestion on some rule-of-thumb?
>
>  Using the methodology proposed in [2], we performed gradient ascent for one epoch on the forget set $S_f$ tuning learning rates as a hyperparameter, consistent with the approach in [2]. Hence, GA terminates after 1 epoch. The comprehensive results of this hyperparameter search are available in the Appendix, with a focus on presenting the strongest results in the main paper.
>
> **Updated related work**
>
> > One related prior work should be included and discussed, see [1].
>
> Thanks for suggesting [3] as related work. The manuscript has been revised accordingly, with [3] now appropriately cited within the related work section.
>
>
> We thank the reviewer for their thoughtful comments and feedback. We hope we addressed all your questions/concerns/comments adequately. In light of our clarifications, please consider increasing your score to accept.
>
> ---
>
> **References**
>
> [1] Nicholas Carlini, Steve Chien, Milad Nasr, Shuang Song, Andreas Terzis, and Florian Tramer. Membership inference attacks from first principles. In 2022 IEEE Symposium on Security and Privacy (SP), pages 1897–1914. IEEE, 2022
>
> [2] Joel Jang, Dongkeun Yoon, Sohee Yang, Sungmin Cha, Moontae Lee, Lajanugen Logeswaran, and Minjoon Seo. Knowledge unlearning for mitigating privacy risks in language models. In Proceedings of the 61st Annual Meeting of the Association for Computational Linguistics (ACL), 2023.
>
> [3] KGA: A General Machine Unlearning Framework Based on Knowledge Gap Alignment.  In Proceedings of the 61st Annual Meeting of the Association for Computational Linguistics (ACL), 2023.

---

### Author Response · Authors · 2023-11-23
**General Comment**

We express our gratitude to all reviewers for their valuable feedback, which has significantly contributed to the refinement of our work. Inspired by their insightful comments, we have implemented the following changes (we have highlighted changes to the paper in green, and relocations of existing text into other sections in blue):


- **Section 3.2 Clarification**: We have revised Section 3.2 to provide a more transparent explanation of our evaluation strategy, addressing detailed questions from Reviewer poKX.

- **Empirical Evidence for Larger Deletion Requests**: In response to the feedback from Reviewers poKX and Awru, we have included additional empirical evidence demonstrating the efficacy of unlearning larger volumes of points using ICUL (see Appendix C.2).

- **Evaluation on Additional LLMs**: Addressing the suggestions from Reviewer Awru, we conducted ICUL experiments on two additional Language Models, Pythia 1B, and Llama2 7B, demonstrating the consistent effectiveness of our approach (see Appendix C.1).

- **Memory Requirement Clarification**: In response to Reviewer fKNk's input, we have clarified the fundamentally lower memory requirement of ICUL compared to Gradient Ascent (GA).

- **Further clarifications**: Responding to Reviewer fKNk, we emphasized that ICUL is particularly beneficial in GPU resource-constrained settings or 'ML as a Service' platforms where only black-box access to the model is available. We also clarified that our work is fundamentally different from [2] which does not study unlearning and instead focuses on the impact of random label flipping on the model's ability to make correct predictions.

We would additionally like to highlight that our work makes several impactful contributions:

- **Black-Box Unlearning for 'ML as a Service'**: We are the first to formulate unlearning in a black-box setting tailored to the practical constraints of 'ML as a Service' platforms. This aspect is particularly crucial in the privacy literature, as neglecting deletion requests may lead to credible claims of copyright infringement when the underlying data is protected by copyright ([1]).

- **In-Context Learning for Machine Unlearning**: Our work pioneers the use of in-context learning for machine unlearning. We specifically construct contexts that induce model behavior at inference time, mimicking the behavior of a re-trained model.

- **Lower Memory Requirements**: Notably, our method boasts lower memory requirements compared to state-of-the-art unlearning methods like GA, especially as the size of Language Models increases. For instance, on Llama2 7B, ICUL can run on a Tesla V100 GPU with 32GB of RAM, while running GA requires access to an A100 GPU with 80GB of RAM. This distinction is particularly crucial, making ICUL computationally feasible for LLMs with billions of parameters.


We appreciated the thorough and insightful review process, which has undoubtedly strengthened the robustness and impact of our work.

---
**References**

[1] Peter Henderson, Xuechen Li, Dan Jurafsky, Tatsunori Hashimoto, Mark A. Lemley, and Percy Liang. "Foundation models and fair use." arXiv:2303.15715, 2023.

[2] Jerry Wei, Jason Wei, Yi Tay, Dustin Tran, Albert Webson, Yifeng Lu, Xinyun Chen, Hanxiao Liu, Da Huang, Denny Zhou, et al. Larger language models do in-context learning differently. arXiv:2303.03846, 2023

---

### Meta-Review · Area_Chair_yiqj · 2023-12-08

**Metareview:**

The proposed approach performs unlearning of specific training instances by providing contextual inputs at the inference stage without needing to modify the model parameters.


**STRENGTHS**

(1) The proposed problem via in-context unlearning is interesting.

(2) The proposed simple approach is cost-efficient relative to gradient-based unlearning methods.


**WEAKNESSES**

The authors have resolved a number of concerns (e.g., experimentation with larger models) raised by the reviewers in their rebuttal. We acknowledge that some of the concerns are challenging to address at this point (e.g., theoretical analysis).

However, a number of major concerns remain and recommendations are provided below for some of them to improve the paper:

(1) The authors have repeatedly highlighted that they situate their work in the blackbox 'ML as a service' platform setting that is practical enough in the real world, at least over gradient-based methods due to computational expense. There is a need to elaborate in greater detail what exactly this setting would be such that it would address several immediate practical concerns.

For example, the authors seem to have missed the question posed by Reviewer Awru: "How is ICUL designed to scale for real-world applications that demand continuous unlearning?" Does it mean that the exemplars to be forgotten keep increasing in size in the prompt? How can we know that all the to-be-forgotten example pairs would be indeed erased since there would be "interactions" between the example pairs to influence the query response?

As another example, since the model is blackbox, how can one tell whether the model has that data to be erased? Are there deterministic means for doing so? If not and unlearning is still performed, wouldn't the label flipping cause the model to predict worse than it should be?


(2) The lack of generalizability of the proposed approach to the multi-class or more complex NLP tasks is a significant concern.


(3) Related to the first concern, though the authors have increased the number of examples to be erased up to 10 in the rebuttal, in the context of machine unlearning at this moment, this seems to be too few. More importantly, do we know the fundamental breaking limit, i.e., the proposed method stops performing well after some threshold?


In summary, while the paper has proposed an interesting simple idea (with a nuanced perspective of the use of label flipping from [2]), there is a need for a more complete treatment in this paper.

The authors are strongly encouraged to revise their paper based on the above feedback and that of the reviewers.

**Justification For Why Not Higher Score:**

A number of major concerns remain, as highlighted in the meta-review.

**Justification For Why Not Lower Score:**

N/A.

---

### Decision · Program_Chairs · 2024-01-16

Reject